# Towards a Unified Generative Model for Scarce Time Series with Domain Experts

**Zihao Yao**[1]  **Qi Zheng**[1]  **Jiankai Zuo**[1]  **Yaying Zhang**[1]

## Abstract

Synthesizing realistic time series with generative models has wide-ranging applications in real-world scenarios. Despite recent progress, most existing methods are trained under the assumption of abundant training data, which substantially limits their effectiveness in data-scarce settings. In this paper, we propose TimeMoDE, a novel framework that integrates Diffusion Transformers with Mixture-of-Experts to exploit both domain adaptability and diffusion-stage awareness for time series generation under data scarcity. It is pre-trained on a large-scale collection of multi-domain datasets to extract domain-agnostic temporal representations and domain-specific information benefiting generalization during fine-tuning. We propose Domain Prompts to condition expert assignment for indistinguishable noised tokens, mitigating the limitations of capturing inter-dataset relationships. Moreover, we incorporate diffusion timestep signals to equip the experts with awareness of time series degradation variations, facilitating adaptive calibrate to stage-dependent denoising requirements. Extensive experiments demonstrate that TimeMoDE outperforms existing methods under diverse low-data settings. It establishes an innovative paradigm for advanced time series few-shot generation.

## 1. Introduction

Time series (TS) data are widely adopted across diverse domains and play a crucial role in numerous applications. However, the collection of high-quality time series is often constrained in both academic research and industrial sectors

---

[1]The Key Laboratory of Embedded System and Service Computing, Ministry of Education, Tongji University, Shanghai, China. Correspondence to: Yaying Zhang <yaying.zhang@tongji.edu.cn>.

*Proceedings of the 43rd International Conference on Machine Learning*, Seoul, South Korea. PMLR 306, 2026. Copyright 2026 by the author(s).

by privacy concerns, limited samples, and high acquisition costs (Gonen et al., 2025). According to scaling laws, insufficient training data may result in model performance bias in downstream tasks (Wang et al., 2025b).

In recent years, generative models that synthesize realistic time series closely resembling original data distribution have emerged as a promising solution (Yao et al., 2025). In particular, denoising diffusion probabilistic models (DDPMs) have become the prevailing generative paradigm (Yuan & Qiao, 2024). Despite the progress, most existing methods are developed and evaluated under the naive presumption of abundant training data within single domain. Consequently, their performance can be constrained in real-world scenarios characterized by data scarcity (Gonen et al., 2025). A feasible approach is to pre-train a general model on large-scale datasets to acquire various temporal representations that improve the generalization of fine-tuned models on downstream tasks. Following this idea, foundation models for time series forecasting and classification have attracted increasing attention (Wang et al., 2024; Shi et al., 2024). However, research on cross-domain time series generation remains relatively limited (Gonen et al., 2025; Huang et al., 2025). This gap may stem from the inherent requirement of cross-domain generation to synthesize diverse time series without access to available records during diffusion process, posing challenges from the following two perspectives.

**Indistinguishable noise hinders effective cross-domain modeling.** Cross-domain modeling requires incorporating domain-specific information to produce time series aligned with target-domain characteristics. In forecasting tasks, input time series from multiple sources display substantial inter-domain heterogeneity, such as trend and periodicity (Wang et al., 2024). In contrast, as illustrated in Figure 1, noise-corrupted time series in diffusion lose intrinsic domain features and become indistinguishable. This prevents model from inferring target domain and consequently leads to synthesized samples that fail to faithfully reflect the desired domain-specific patterns. A straightforward approach involves class labels as distinguishment (Gonen et al., 2025). Nevertheless, this strategy relies on the availability of predefined classes and struggle to generalize to previously unseen domains. Besides, one-hot encoding of class labels implic-

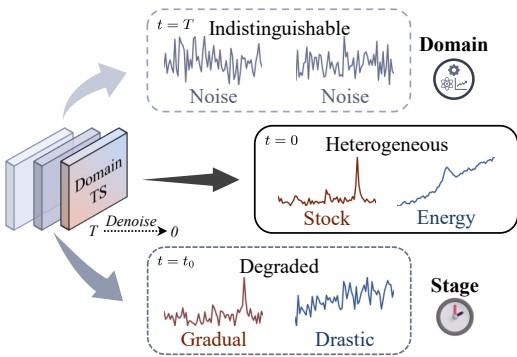

*Figure 1.* The challenges arise from cross-domain generation. First, multi-source time series ($t = 0$) are corrupted into indistinguishable noise ($t = T$) that hinders domain identification. Second, heterogeneous time series exhibit distinct degradation that obscures the semantic information of diffusion timesteps ($t = t_0$).

itly assumes that datasets are mutually independent which neglects their potential relationships. Another alternative is to leverage natural language descriptions to discern data sources (Liu et al., 2024). However, modality discrepancies hinder precise nuance preservation, ultimately leading to incomplete and ambiguous prompts (Huang et al., 2025).

**Variations of time series degradation obscure the semantics of diffusion timesteps.** Generative models exhibit stage-dependent denoising requirements during diffusion process. Specifically, the early stages focus on high-level structures, while the later stages emphasize the recovery of fine-grained details (Cheng et al., 2025a). However, heterogeneous time series exhibit distinct degradation under the same noise schedule (Lee et al., 2024), hindering models from precisely inferring current stage from input representations. As shown in Figure 1, under the same diffusion timestep, the Stock time series undergo gradual corruption and retain much of informative content, whereas the Energy time series is drastically corrupted and approach a noise-like state. The monolithic design of existing methods (Gonen et al., 2025; Huang et al., 2025), which densely activate parameters and treat all noised time series uniformly, can lead to either excessive denoising that destroys local details or insufficient denoising that produces meaningless outputs.

To tackle these challenges, we propose a unified generative framework for time series under data scarcity with domain experts, namely TimeMoDE. Unlike conventional models that are trained separately for each dataset and require tailored parameter tuning, TimeMoDE is pre-trained on a large-scale collection of datasets spanning multiple domains to learn generalized temporal representations and transferable domain knowledge, supporting effective fine-tuning in low-data regimes. Motivated by the recent achievements of Mixture-of-Experts (MoE) in time series analysis (Liu et al., 2025; Sun et al., 2024), we pioneer the exploration of this

architecture within diffusion-based frameworks for cross-domain time series generation. Considering the indistinguishability of noised inputs and the limitations of class labels, we propose Domain Prompts and time series prototypes to condition expert assignment. Through training, the prototypes progressively refine subspace basis to identify the source domains described by prompts under time series semantics. During fine-tuning, previously unseen samples can adaptively select experts associated with time series exhibiting similar temporal patterns during pre-training, facilitating estimation of latent distribution. Moreover, we introduce diffusion timestep conditioning into experts to enhance the diffusion stage awareness and representational capacity of TimeMoDE. It empowers experts to infer the current stage and dynamically adapt to distinct denoising demands. By jointly designing the routing mechanism and expert networks, TimeMoDE achieves specialized processing with respect to both domain content and diffusion-stage context. Our contributions are summarized as follows:

- We propose TimeMoDE, a pioneering general time series generation model pre-trained across diverse domains, empowering generalization capability and fine-tuning performance under data scarcity.

- We develop a novel routing mechanism that assigns indistinguishable noise to relevant experts for domain-specialized processing. We further enhance experts with diffusion timestep awareness to flexibly adjust stage-dependent denoising requirements.

- Extensive experiments on multi-domain real-world datasets demonstrate the superiority of TimeMoDE in various data-scarce scenarios. The empirical results and analyses provide insights for future research.

## 2. Related Work

### 2.1. Time Series Generation

Time series generation models can be categorized into three main paradigms. The first branch is based on Generative Adversarial Networks (GANs) (Goodfellow et al., 2014; Yoon et al., 2019), which jointly optimize generator and discriminator to generate realistic temporal dynamics. The second branch is Variational Autoencoder (VAEs)-based methods (Desai et al., 2021; Naiman et al., 2024b) that leverage specific decoder for latent temporal representations. Recently, the effectiveness of DDPMs in images and natural language (Peebles & Xie, 2023; Fei et al., 2024) has promoted their development in time series generation (Li et al., 2025; Huang et al., 2025; Yao et al., 2025). Despite the progress, most existing methods are designed within a single-domain with the naive assumption of abundant training data. Inspired by (Gonen et al., 2025) that emphasizes the need for gen-

eration feasibility under low-data conditions, we propose TimeMoDE to tackle the challenge in this work.

## 2.2. Time Series Foundation Model

Foundation models (FM) aim to achieve generalization ability and superior performance with minimal fine-tuning across different tasks by pre-training on large-scale datasets (Bommasani, 2021). One stream of work explores adapting language models for time series to capture sequence dependencies (Liu et al., 2024; Cheng et al., 2025b). Another stream is dedicated to designing tailored architectures to handle heterogeneous time series across different domains (Das et al., 2024; Jiang et al., 2025). Recently, FM incorporating Mixture-of-Experts (MoE) that sparsely activates sub-networks for distinct inputs has garnered widespread attention (Sun et al., 2024; Cheng et al., 2025a). Time-MoE (Shi et al., 2024) attempts to scale to billions of parameters via MoE and attains improved forecasting precision. Moirai-MoE (Liu et al., 2025) introduces a new gating function to enable token-level specialization. Despite advances in forecasting task, the potential of MoE for time series generation remains largely unexplored, primarily hindered by the difficulty of assigning experts to indistinguishable noise. Limited data from unseen datasets further exacerbates this issue. In this work, we propose Domain Prompts and prototypes to adaptively assign domain experts, unlocking the power of MoE in cross-domain time series generation.

## 3. Preliminaries

### 3.1. Problem Statement

Let $\mathcal{D} = \{x^{(i)}\}_{i=1}^{M}$ denotes a time series dataset with $M$ samples. Each sample $x^{(i)} \in \mathbb{R}^{H \times C}$ is a multivariate time series drawn from an unknown distribution $p_0(x)$, where $H$ is the sequence length and $C$ denotes the number of channels. $M$ is typically small in the few-shot setting, which makes it challenging for generative models to estimate $p_0(x)$ from limited samples. It simulates the data-scarce scenarios encountered in practice. The goal is to construct a generative model $p_\theta(x)$ from restricted dataset $\mathcal{D}$ to map Gaussian noise to sufficient time series, which approximates the target distribution such that $p_\theta(x) \approx p_0(x)$.

### 3.2. Denoising Diffusion Probabilistic Models

Diffusion models are a type of generative model that learn to generate data by gradually reversing a stochastic noise process. In the forward Markov process, samples $x_0 \sim q(x)$ are gradually corrupted by noise at each diffusion step $t$ into standard Gaussian noise $x_T \sim \mathcal{N}(0, \mathbf{I})$:

$$q\left(\mathbf{x}_t \mid \mathbf{x}_{t-1}\right) = \mathcal{N}\left(\mathbf{x}_t; \sqrt{1 - \beta_t}\mathbf{x}_{t-1}, \beta_t\mathbf{I}\right), \quad (1)$$

where $t \in [1, T]$, $\beta_t \in (0, 1)$ defines the noise schedule. The reverse process denoise samples via reverse transitions:

$$p_\theta\left(\mathbf{x}_{t-1} \mid \mathbf{x}_t\right) = \mathcal{N}\left(\mathbf{x}_{t-1}; \mu_\theta\left(\mathbf{x}_t, t\right), \Sigma_\theta\left(\mathbf{x}_t, t\right)\right), \quad (2)$$

where $\mu_\theta(\cdot)$ is a learnable parameter, $\sum_\theta(\cdot)$ is fixed as constant depending on $\beta_t$. The reverse process can be reduced to training a noise predictor $\epsilon_\theta$ to parameterize $\mu_\theta(\cdot)$ at each step $t$. The parameters $\theta$ are optimized by minimizing:

$$\mathcal{L}_{\text{DDPM}}(\theta) = \mathbb{E}_{x_0, \epsilon, t}[\lambda(t)\|\epsilon - \epsilon_\theta(x_t, t)\|^2], \quad (3)$$

where $\lambda(t)$ is a weight that changes noise scale.

## 4. Methodology

In this section, we introduce TimeMoDE, a novel framework that exploits both domain adaptability and diffusion-stage awareness for time series generation under low-data settings. We first describe the architectural design that enhances the Diffusion Transformer (DiT) (Peebles & Xie, 2023) with well-established Mixture of Domain Experts (MoDE) module. Then, we propose the routing mechanism that assigns tokens based on Domain Prompts (DP) and prototypes, inducing domain-specialized modeling while facilitating knowledge transfer across related datasets. Furthermore, we introduce an augmented expert network that explicitly accounts for varying degradation of time series, enabling dynamic adjustment of expert behavior throughout diffusion process. Finally, we present the two-stage pre-training and fine-tuning protocol that strengthen the capability of cross-domain few-shot time series generation.

### 4.1. Model Framework

Since DiT (Peebles & Xie, 2023) has been validated as an effective diffusion framework for high throughput and condition incorporating, we build TimeMoDE on the backbone of DiT as depicted in Figures 2. As single time point contains limited information, we employ prevalent patching technique to effectively capture local semantic information (Liu et al., 2025). Given a noised time series input of length $H$, we segment it into non-overlapping patches of length $S$, resulting in $N = \lceil \frac{H}{S} \rceil$ patches $z_i \in \mathbb{R}^{S \times C}$. These patches are then embedded into a dimension $d$ via linear transformation (Li et al., 2025) to learn contextual representations:

$$z_{\text{PE}_i} = \text{Emb}(z_i) + \text{PE}_i, \quad (4)$$

where $\text{PE}_i$ denotes the learned positional embedding, and $z_0 = \{z_{\text{PE}_1}, \ldots, z_{\text{PE}_N}\} \in \mathbb{R}^{N \times d}$ represents the enriched patch representations. The diffusion timestep $t$ and class label are jointly embedded into condition $c \in \mathbb{R}^d$ via a feedforward network (FFN).

To enhance representational expressiveness, we introduce DiT-MoDE composed of $L$ stacked layers. It replaces the

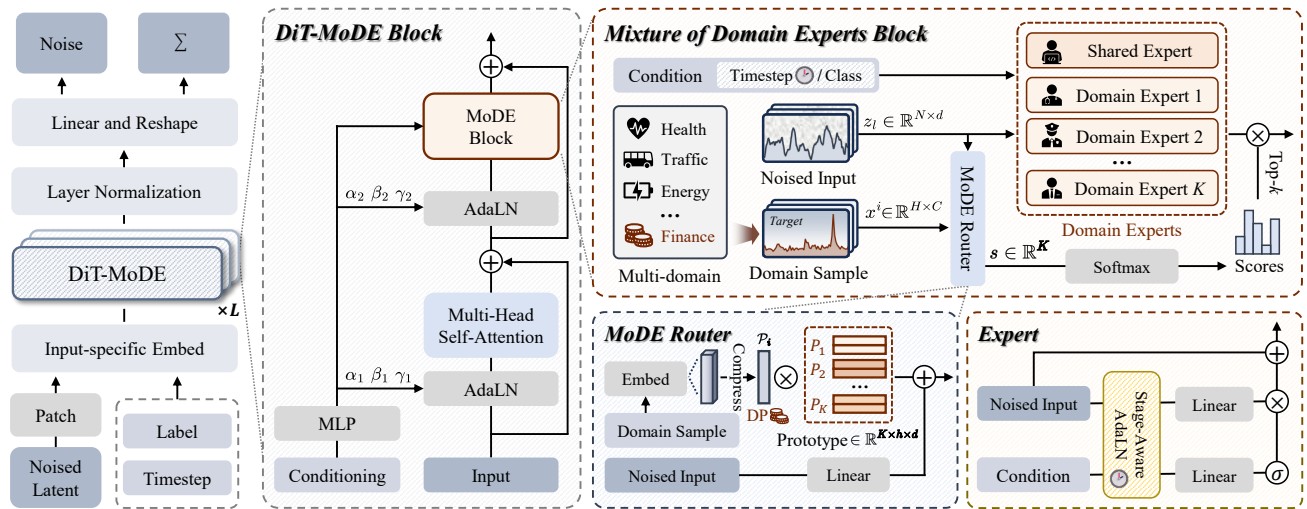

*Figure 2.* Proposed TimeMoDE framework. TimeMoDE achieves domain adaptability and diffusion-stage awareness for time series generation by replacing conventional MLPs in Diffusion Transformer with a carefully designed Mixture of Domain Experts (MoDE).

vanilla MLPs in DiT with MoDE, where each sub-network shares an identical architecture with embedded lightweight stage signals. Within this modular design, experts are sparsely activated according to scores computed by routing function. The process can be formally expressed as:

$$
\begin{aligned}
z_l &= \text{MHSA}(\text{AdaLN}(z_l, c)) + z_l, \\
z_{l+1} &= \text{MoDE}(\text{AdaLN}(z_l, c)) + z_l,
\end{aligned}
\tag{5}
$$

where $\text{MHSA}(\cdot)$ represents the Multi-Head Self-Attention, $z_l$ is the output of the $l$-th DiT-MoDE layer.

## 4.2. MoDE Router

### 4.2.1. DOMAIN PROMPT

The entanglement of domain characteristics and noise factors complicates optimization, as model struggles to disentangle whether a token representation is driven by domain semantics or noise (Cheng et al., 2025a). Previous work (Gonen et al., 2025) attempts to inject class labels as guidance to disentangle domain effects. However, it proves ineffective when encountering new domains, as it assumes that datasets are entirely independent which neglects inter-domain similarity and potential shared patterns. While natural language descriptions are effective in NLP, they struggle to capture comprehensive temporal features due to modality discrepancies. To this end, we propose Domain Prompts (DP) that serve as textual cues within the context of time series.

In the pre-training stage, we randomly sample instances from training set as representative domain exemplars. These samples are assumed to approximate underlying distribution, thereby implicitly describing the temporal characteristics of corresponding domain. Each sample is then encoded into a hidden representation $\mathcal{P} \in \mathbb{R}^d$ to construct DP:

$$
\mathcal{P} = \text{Avg}(\text{Linear}(\text{Conv}(x)) + \text{PE}),
\tag{6}
$$

where $\text{Conv}(\cdot)$ refers to dynamic convolution along channels to capture multivariate correlations, $\text{Avg}(\cdot)$ denotes temporal pooling. It compresses original time series with dispersed channel and temporal information into a compact global representation that serves as domain identity.

During fine-tuning and inference on new datasets, we construct DP using the available $M$ few-shot samples. If the expected number of generated time series exceeds $M$, these samples are repeatedly utilized to construct DP until the desired sample count is reached.

### 4.2.2. TIME SERIES PROTOTYPE

To accurately determine the domain of input, we equip each expert with prototype to identify the property of DP. During pre-training, the learnable basis within prototype is progressively refined to store valuable prior knowledge. In the fine-tuning phase, the basis autonomously matches DP to the most relevant experts pre-trained on similar time series, ultimately generating samples that adhere to the target domain while not being constrained by the limited temporal patterns present in scarce data.

Specifically, we initialize set $P = \{P_1, P_2, \ldots, P_K\} \in \mathbb{R}^{K \times h \times d}$ consisting of $K$ prototypes, where each $P_i$ is viewed as a representation subspace of specific domain spanned by $h$ basis. Given that time series may exhibit evolving trend and periodicity leading to non-stationarity within the same domain, a single basis vector may cause model to focus only on coarse-grained patterns. To ensure that the subspace covers the full spectrum of features required to precisely identifying DP, we set up multiple basis

and normalize them as $\|P_i^j\| = 1$, where $i \in \{1, \ldots, K\}$ and $j \in \{1, \ldots, h\}$. We aim to maximize the differentiation among the basis within each prototype, such that a minimal number of basis can span a comprehensive subspace that effectively captures various patterns. Accordingly, we impose a constraint that encourages orthogonality among basis:

$$\mathcal{L}_{(i)} = \|P_i P_i^\top \odot \mathbf{I} - \mathbf{I}\|_F^2, \tag{7}$$

where $\odot$ is the Hadamard product, $\mathbf{I} \in \mathbb{R}^{d \times d}$ denotes an identity matrix. The total loss over $K$ prototypes is computed as $\mathcal{L}_{\text{basis}} = \sum_{i=1}^{K} \mathcal{L}_{(i)}$.

In a similar manner, we seek to maximize the separability among $K$ prototypes so that each subspace focuses on temporal attributes within a specific domain. The constraint between pairs of domains can be expressed as:

$$\mathcal{L}_{i,j} = \|P_i P_j^\top\|_F^2, \quad i, j \in \{1, \ldots, K\} \text{ and } i \neq j. \tag{8}$$

The loss for all possible pairwise subspaces is given by:

$$\mathcal{L}_{\text{sub}} = \sum_{i \neq j} \|P_i P_j^\top\|_F^2 = \|PP^\top \odot \mathbf{B}\|_F^2. \tag{9}$$

$\mathbf{B}$ denotes the block diagonal matrix, where each diagonal block of $h$-size equals to 0, with all other elements set to 1. These two constraints can be integrated into a prototype-based loss, $\mathcal{L}_{\text{proto}}$, which maximize both intra-prototype diversity and inter-prototype separability.

We measure the matching degree between DP $\mathcal{P}_j$ and prototype $P_i$ using cosine similarity. Since each input contains its own individual information beyond population-level characteristics, we incorporate a linear layer to calibrate assignment scores based on input. Hence, the probability that $z_l$ belongs to the $i$-th expert can be formulated as:

$$s_i = \frac{\|\mathcal{P}_j P_i\|_F^2 + W z_l[i] + \tau}{\sum_i \|\mathcal{P}_j P_i\|_F^2 + W z_l + K\tau}, \tag{10}$$

where $\tau$ is a temperature parameter that controls the smoothness, $W \in \mathbb{R}^{K \times d}$ denotes learnable weight matrix.

### 4.3. Domain Experts

#### 4.3.1. EXPERT NETWORKS

MoE-based time series forecasting models (Liu et al., 2025; Shi et al., 2024; Sun et al., 2024) typically employ static expert networks, such as MLPs, which are limited in diffusion modeling since they treat all timesteps uniformly and overlook the varying noise intensities and denoising demands. Moreover, heterogeneous time series may display distinct non-stationarity and attenuation of frequency components (Wang et al., 2025a) under the same noise schedule and diffusion timestep, which confounds the semantics encoded in the shared timestep embeddings of DiT.

Drawing inspiration from Adaptive Layer Normalization (AdaLN) (Peebles & Xie, 2023; Cheng et al., 2025a), we augment expert with lightweight timestep signals, enabling dynamic adaptation to domain-specific diffusion stages. In practice, each input is normalized and then adaptively scale and shift using learnable parameters conditioned on diffusion timestep and class label embedding: $z_l = \text{AdaLN}(z_l, c)$. It encourages experts to focus on relevant features at current stage. The stage-aware representations are then passed to gated linear units (GLU):

$$z_l' = W_{\text{down}}(\text{SiLU}(W_{\text{gate}} z_l) \odot W_{\text{up}} z_l) + z_l, \tag{11}$$

where $W_{\text{down}} \in \mathbb{R}^{nd \times d}$, $W_{\text{gate}}, W_{\text{up}} \in \mathbb{R}^{d \times nd}$ are learnable weight matrices, $\text{SiLU}(\cdot)$ is the nonlinear activation function, and $n$ is the expansion ratio of expert latent space.

#### 4.3.2. OUTPUT AGGREGATION

Unlike existing generative methods (Gonen et al., 2025) that employ uniform distribution modeling with dense parameter activation, TimeMoDE sparsely activates a subset of experts relevant to each token to promote domain-specialized modeling and distinctive information incorporation.

MoDE contains $K$ experts denoted as $\{E_1, E_2, \ldots, E_K\}$. Notably, we do not explicitly associate a dedicated expert with each domain, as domain boundaries across datasets are ambiguous and previously unseen datasets may not correspond to known domains. In fact, different individual time series are assumed to share a common set of components but reflect distinct subsets of the collection (Huang et al., 2025). While experts capture overlapping or complementary structures, their weighted aggregation can emphasize certain features or produce new representations, supporting diverse domain-specialized behaviors without being constrained by the finite number of experts. We compute the weights of experts via gating function based on assigned scores:

$$G(s_i) = \text{Softmax}(\text{Top-}k(s)), \tag{12}$$

where $k$ denotes the predefined number of selected experts. Besides, an additional expert $E_0$ is designated as a shared expert to capture and consolidate common knowledge across different contexts (Shi et al., 2024). Thus, the final output of the $l$-th MoDE layer is computed as:

$$z_l' = \sum_{i=1}^{K} G(s_i) \cdot E_i(z_l) + E_0(z_l). \tag{13}$$

To encourage full utilization of parameters rather than repeatedly activating only a few experts known as the load balancing issue (Shazeer et al., 2017), an auxiliary load balancing loss is applied within a batch (Liu et al., 2025):

$$\mathcal{L}_{\text{aux}} = \sum_{i=1}^{K} \left( \frac{1}{\mathcal{B}} \sum_{j=1}^{\mathcal{B}} \mathbb{1}_{c_i} \right) \cdot \left( \frac{1}{\mathcal{B}} \sum_{j=1}^{\mathcal{B}} g_i \right), \tag{14}$$

where $\mathcal{B}$ is the total number of token in batch, $\mathbb{1}_{c_i}$ denotes the indicator function that the token selects expert $i$ and $g_i$ is the weight distribution of token allocated to expert $i$.

## 4.4. Training Protocol

### 4.4.1. PRE-TRAINING

We pre-train TimeMoDE on time series collection spanning a wide range of statistical properties. The unified architecture is expected to capture both domain-specific temporal dynamics and domain-agnostic transferable representations that benefit downstream tasks. To better identify domain and assign corresponding experts, in addition to diffusion loss that predicts original time series from noise (Eq. 3), we incorporate the aforementioned prototype-based loss and an auxiliary assignment loss during the pre-training stage:

$$\mathcal{L} = \mathcal{L}_{\text{DDPM}} + \mathcal{L}_{\text{proto}} + \mathcal{L}_{\text{aux}}. \tag{15}$$

### 4.4.2. FINE-TUNING

TimeMoDE is then fine-tuned on a previously unseen dataset with limited instances to simulate data-scarce scenarios. This challenging task requires model to rapidly adapt to low-resource regimes while estimating underlying distribution.

The basis within prototypes store domain information acquired from pre-training. During fine-tuning, they are frozen to preserve domain semantics, and the prototype-based loss is accordingly deactivated. Besides, only experts pertinent to target domain are activated, since engaging all parameters may introduce extraneous information from other domains. Consequently, the load balancing loss is also removed.

## 5. Experiments

### 5.1. Experiment Settings

**Dataset.** The raw data used for pre-training and fine-tuning are collected from a variety of public time series datasets (Gonen et al., 2025; Gao et al., 2024) spanning 10 domains—healthcare, finance, energy, traffic, cloud, human activity, machine sensors, physics, space, and nature—and covering 5 tasks: forecasting, simulation, anomaly detection, classification, and generation. We simulate data-scarce scenarios by restricting training to a small fraction of dataset. Specifically, the few-shot training ratios are set to $5\%$, $10\%$, and $15\%$. We also investigate an extreme low-resource setting in which only 10, 25, and 50 instances are available. These configurations comprehensively cover a range of data-scarcity scenarios and provide a thorough evaluation.

**Evaluation.** We carefully select several representative and competitive models from recent researches as baselines, including ImagenFew (Gonen et al., 2025), ImagenTime (Naiman et al., 2024a), DiffusionTS (Yuan & Qiao, 2024),

and KoVAE (Naiman et al., 2024b). To comprehensively quantify the quality of synthesized data, including distributional diversity, fidelity of temporal dependencies, and practical usefulness in downstream applications, we employ the mainstream evaluation metrics (Yao et al., 2025), including Context-Fréchet Inception Distance score (c-FID), Discriminative Score (Disc.), and Predictive Score (Pred.).

**Implementation details.** We pre-train TimeMoDE for 1000 epochs using AdamW optimizer with a learning rate 1e-4. All experiments are conducted on a machine with NVIDIA V100 GPU and 32 GB memory. We apply exponential moving average (EMA) with a decay factor 0.9999 to parameters during training to enhance overall stability. Additional details on experimental setup are presented in Appendix B.

### 5.2. Main Results

**Few-shot generation.** We report the averaged metric values for 24-length time series generation in Table 1. Among all baselines concerned, TimeMoDE achieves the overall best performance under both percentage-based and count-based low-data settings. Compared with models trained on limited samples from a single domain, which typically suffer from substantial performance drop, TimeMoDE leverages abundant pre-training data to learn latent temporal dependencies and distribution, enabling effective reconstruction on new dataset using only a few examples. In comparison with ImagenFew adhering pre-training and fine-tuning protocol, TimeMoDE demonstrates consistent competitive performance. While the class label of ImagenFew offers little benefit when generalizing to unseen domains, our proposed DP can capture inter-domain differences and facilitate adaptive expert routing. Quantitatively, TimeMoDE achieves an average reduction of approximately $25\%$ in c-FID and an improvement of over $30\%$ in Discriminative Score.

**Full-shot generation.** TimeMoDE is also validated under data-rich settings. As shown in Table 2, TimeMoDE continues to benefit from increased available data and outperforms advanced methods. Notably, TimeMoDE fine-tuned with only $5\%$ of dataset even surpasses state-of-the-art models such as ImagenTime trained on full dataset. It highlights the effectiveness of transferring knowledge from broad datasets and demonstrate the general applicability of our model.

**Visualization.** We employ t-SNE and kernel density estimation (Yuan & Qiao, 2024) to provide an intuitive understanding of model behavior in Figure 3. As shown, TimeMoDE simultaneously captures global structures and replicates local features, resulting in closer alignment with the original data, whereas other methods either fail to achieve comprehensive coverage or to capture underlying distribution.

**Additional metrics.** To thoroughly evaluate the diversity of synthesized data and the fidelity to original characteristics,

*Table 1.* Averaged few-shot generation results in terms of context-FID (c-FID ↓), Discriminative Score (Disc. ↓) and Predictive Score (Pred. ↓) across subset scales. A lower value indicates a better performance. The best scores are in bold and the second best are underlined.

| Subset size | Metric | TimeMoDE | ImagenFew | ImagenTime | DiffusionTS | KoVAE | Improvement |
|---|---|---|---|---|---|---|---|
| 5% | c-FID | **0.355** | 0.426 | 2.691 | 3.004 | 3.076 | 16.655% |
| | Disc. | **0.100** | 0.169 | 0.289 | 0.322 | 0.321 | 40.936% |
| | Pred. | **0.533** | 0.537 | 0.547 | 0.584 | 0.571 | 0.856% |
| 10% | c-FID | **0.240** | 0.334 | 2.084 | 2.756 | 2.613 | 28.109% |
| | Disc. | **0.078** | 0.126 | 0.222 | 0.298 | 0.283 | 37.739% |
| | Pred. | **0.532** | 0.535 | 0.544 | 0.580 | 0.563 | 0.673% |
| 15% | c-FID | **0.184** | 0.285 | 1.885 | 1.922 | 1.535 | 35.499% |
| | Disc. | **0.064** | 0.079 | 0.206 | 0.301 | 0.228 | 19.187% |
| | Pred. | **0.531** | 0.534 | 0.542 | 0.563 | 0.556 | 0.506% |
| #10 | c-FID | **1.950** | 2.299 | 5.355 | 4.782 | 5.026 | 15.210% |
| | Disc. | **0.264** | 0.304 | 0.352 | 0.376 | 0.392 | 13.114% |
| | Pred. | 0.582 | **0.579** | 0.586 | 0.634 | 0.597 | -0.553% |
| #25 | c-FID | **1.029** | 1.393 | 4.131 | 3.361 | 4.507 | 26.111% |
| | Disc. | **0.183** | 0.267 | 0.339 | 0.376 | 0.367 | 31.510% |
| | Pred. | **0.554** | 0.560 | 0.560 | 0.608 | 0.588 | 1.072% |
| #50 | c-FID | **0.556** | 0.791 | 3.295 | 2.384 | 4.241 | 29.677% |
| | Disc. | **0.132** | 0.240 | 0.316 | 0.334 | 0.358 | 44.875% |
| | Pred. | **0.545** | 0.547 | 0.553 | 0.581 | 0.599 | 0.384% |

*Table 2.* Averaged full-shot generation results.

| Models | c-FID | Disc. | Pred. |
|---|---|---|---|
| TimeMoDE | **0.040** | **0.034** | **0.530** |
| ImagenFew | 0.208 | 0.067 | 0.532 |
| ImagenTime | 0.460 | 0.106 | 0.535 |
| DiffusionTS | 1.696 | 0.259 | 0.555 |
| KoVAE | 0.900 | 0.166 | 0.545 |

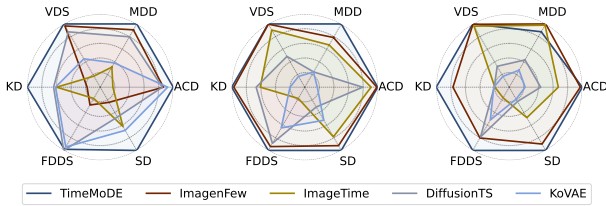

*Figure 4.* Feature-based and population-level metrics comparison on Mujoco (left), ETTh2 (mid), and ILI (right).

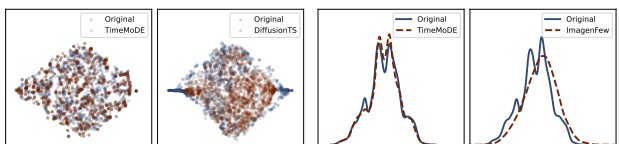

*Figure 3.* t-SNE plots (ETTh2, left) and value distributions (Mujoco, right) on original (red) and synthetic (blue) time series.

we include feature-based metrics summarized in (Ang et al., 2023) and population-level metrics from (Li et al., 2025). Detailed formulas are provided in Appendix B.4. As illustrated in Figure 4, results obtained under 5% training data regime confirm that TimeMoDE preserves key temporal properties during generation across diverse domains.

### 5.3. Ablation Study

We design the following variants to validate the effectiveness and necessity of proposed components: (1) w/o MoDE Router: TimeMoDE assigns experts to each token based solely on input gating function. (2) w/ Dense Layer: This variant replaces MoDE with dense structure of expert, which is similar to DiT. (3) From Scratch: TimeMoDE is trained directly on few-shot target data without pre-training. (4) Timestep Uncond.: The expert networks are replaced with standard MLP layers. (5) Prototype Finetuning: The parameters in prototypes are not frozen during fine-tuning.

As evidenced by Table 3, w/ Dense Layer leads to a consistent and substantial drop across all metrics, underscoring the validity of MoE architecture for time series generation. The uniform modeling manner that densely activates parameters hinders the capture of domain-specific features, ultimately resulting in inferior generation performance. Further performance gains stem from the pre-training datasets, which provides sufficient temporal patterns and shared domain characteristics, enabling TimeMoDE to infer latent distributions from only a few samples. The w/o MoDE Router variant struggles to discern inter-domain correlations from highly corrupted inputs, which obstructs accurate expert assignment for unseen datasets. While incorporating timestep

*Table 3.* Ablation study on key components of TimeMoDE.

| Subset size | Variant | c-FID | Disc. | Pred. |
|---|---|---|---|---|
| 10% | TimeMoDE | **0.240** | **0.078** | **0.532** |
| | w/o MoDE Router | 0.302 | 0.102 | 0.533 |
| | w/ Dense Layer | 0.875 | 0.211 | 0.559 |
| | From Scratch | 0.882 | 0.167 | 0.537 |
| | Timestep Uncond. | 0.244 | 0.097 | 0.532 |
| | Prototype Finetuning | 0.242 | 0.090 | 0.532 |
| #25 | TimeMoDE | **1.029** | **0.183** | **0.554** |
| | w/o MoDE Router | 1.235 | 0.207 | 0.567 |
| | w/ Dense Layer | 1.438 | 0.252 | 0.557 |
| | From Scratch | 1.525 | 0.241 | 0.559 |
| | Timestep Uncond. | 1.053 | 0.198 | 0.557 |
| | Prototype Finetuning | 1.055 | 0.190 | 0.554 |

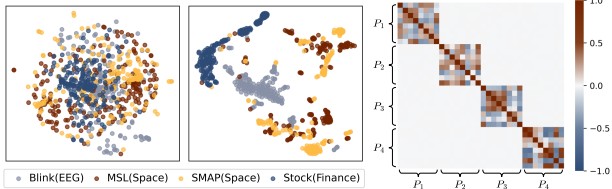

(a) t-SNE of input and DP.  (b) Correlation heatmap.

*Figure 5.* Visualizations of DP and prototypes. (a) Comparison between input and corresponding DP across domains. (b) Cosine similarity among basis within prototypes.

condition into experts yields moderate gain, it enhances the awareness of heterogeneous degradation phase and facilitate fine-grained refinement at later diffusion stage. For full experiment results, please refer to Appendix C.3.

### 5.4. MoDE Analyses

**Domain Prompts.** To validate the superiority of DP over conventional input-based gating function, we employ t-SNE for comparison. In the first column of Figure 5a, input embeddings from different domains are heavily mixed despite distinct temporal patterns. In contrast, DP in the second column shows clear clustering modes. Representations from Blink (EEG) and Stock (Finance) are dissimilar and well separated, whereas MSL and SMAP, both from Space, are successfully blended. This confirms the capability of TimeMoDE to extract relationships among datasets.

**Prototypes.** We visualize the cosine similarity among basis within prototypes. As shown in Figure 5b, owing to the design of loss constraints, prototypes of different experts exhibit minimal correlation. It facilitates effective expert assignment and specialized modeling for heterogeneous time series. Meanwhile, basis within the same prototype display pronounced correlations, enabling them to collaboratively capture the full spectrum of features for certain domain.

**Expert assignment.** To investigate the behavior of expert

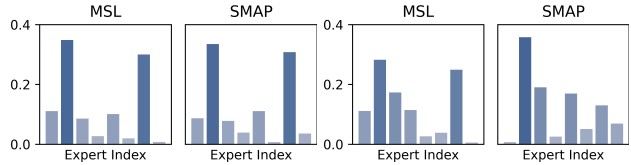

*(a)* Expert assignment in layer 1. *(b)* Expert assignment in layer 6.

*Figure 6.* Assigned expert distribution in shallow and deep layers.

assignment, we analyze activation frequency of MSL and SMAP, both from Space, in shallow (layer 1) and deep (layer 6) layers of DiT-MoDE block. As presented by the normalized activation percentage in Figure 6, their expert allocations are concentrated and nearly identical in the shallow layer, which is expected since datasets originating from the same domain share similar representations. By the final layer, expert selection becomes notably more diverse. This divergence may arises from the dominance of noise in initial state. At shallow layers, TimeMoDE primarily captures generalizable structures interpreted as domain-level features for rapid denoising. As tokens are aggregated in deeper layers, it gradually shifts focus to recovering fine-grained local details, which may reflect shared patterns across domains while exhibiting differences within the same field. This behavior is similar to phenomena observed in LLMs (Zhu et al., 2024), where earlier layers typically concentrate on limited experts to capture common linguistic features, while deeper layers become more diverse depending on tasks.

### 5.5. Discussion

As a unified generative model, TimeMoDE requires pretraining on multiple datasets to fully leverage its generation capability under data scarcity. It incurs additional computational overhead compared to single-domain models. Distributed training on two NVIDIA V100 GPUs takes roughly 20 hours, while fine-tuning on 50 samples with available checkpoint requires only about 4 minutes on a single GPU comparable to ImagenFew around 3 minutes. Given the modest time demands in practice and superior performance in Table 1 and 2, this trade-off is deemed justified.

## 6. Conclusion

In this paper, we present TimeMoDE, a novel unified framework that exploits domain adaptability and diffusion-stage awareness for time series generation under data scarcity. As a key contribution, we propose the routing mechanism that precisely assigns experts relevant to indistinguishable noise to transfer domain-specific information. Moreover, we augment experts with lightweight timestep signals, equipping TimeMoDE with awareness of time series degradation discrepancy to dynamically adapt to stage-dependent denoising requirements. Extensive experiments demonstrate

that TimeMoDE outperforms existing methods under both low-data and full-shot conditions. In-depth analysis further validates the effectiveness of our design and offers insights into time series generative foundation models.

## Acknowledgments

This work was partly supported by the National Key Research and Development Program of China under Grant 2023YFB3002201, the National Natural Science Foundation of China under Grant 72342026, and Fundamental Research Funds for the Central Universities under Grant 2024-6-ZD-02.

## Impact Statement

This paper presents work whose goal is to advance the field of Machine Learning. There are many potential societal consequences of our work, none which we feel must be specifically highlighted here.

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

## A. Detailed Architecture of TimeMoDE

We illustrate the detailed architecture of TimeMoDE in Figure 7. As DiT has been extensively validated as an effective diffusion framework with high throughput and robust conditioning, we adopt it as the backbone of TimeMoDE. To better identify the domain of each input and assign the corresponding experts for domain-specialized modeling, we incorporate the standard diffusion loss $\mathcal{L}_{\text{DDPM}}$, which predicts the original time series from noise, with two additional objectives: a prototype-based loss $\mathcal{L}_{\text{proto}}$ that enforces constraints on subspace basis across prototypes, and an auxiliary assignment loss $\mathcal{L}_{\text{aux}}$ to balance expert loading during pre-training. Since the prototypes have encoded domain-specific features from the pre-training data, and only experts relevant to the downstream task are expected to be activated, we freeze the prototype-related parameters and update the remaining components using only the diffusion loss $\mathcal{L}_{\text{DDPM}}$ term during fine-tuning.

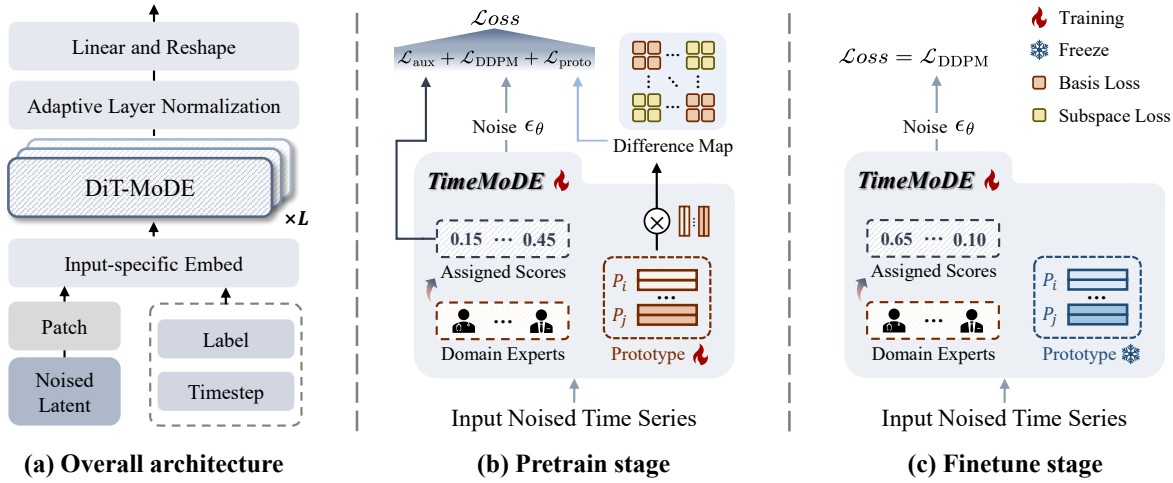

(a) Overall architecture      (b) Pretrain stage      (c) Finetune stage

*Figure 7.* (a) shows the overall framework of TimeMoDE, while (b) and (c) depict the distinct procedures and objectives under the pre-training and fine-tuning protocols, respectively.

## B. Experimental Details

### B.1. Pre-training Datasets

We adopt a two-stage paradigm of pre-training and fine-tuning to fully realize the potential of TimeMoDE, leveraging knowledge acquired from pre-training to infer the latent distribution of limited time series. Consequently, it is essential that the pre-training data cover a sufficiently diverse set of domains and contain an adequate number of samples. Following (Gonen et al., 2025), we employ multi-source datasets originating from a wide range of domains, including healthcare, finance, energy, traffic, cloud, human activity, machine sensors, physics, space, and nature, and spanning various downstream tasks, including forecasting, simulation, anomaly detection, classification, and generation. These datasets exhibit inherent differences in sample size, number of variables, sampling frequency, and other statistical characteristics, as summarized in Table 4. Collectively, they provide broad coverage of real-world scenarios.

### B.2. Evaluation Datasets

We conduct few-shot generation experiments to simulate data-scarce scenarios encountered in real-world applications. Specifically, the pre-trained model checkpoints are retained and subsequently fine-tuned on subsets of target datasets. The subset size is determined either by a predefined percentage or by a fixed number of samples randomly selected from each dataset. To comprehensively evaluate the effectiveness of models, the fine-tuning datasets span diverse domains. These include domains that partially overlap with those seen during pre-training, as well as related domains and entirely unseen domains. A summary of the statistical characteristics of the evaluation datasets is provided in Table 5.

### B.3. Hyperparameter Setting

The hyperparameters of TimeMoDE can be categorized into three groups: hyperparameters specific to the pre-training phase, hyperparameters specific to the fine-tuning stage, and the core architectural parameters shared across both phases. These

*Table 4.* List of pretraining datasets.

| Domain | Dataset | Samples | Variables | Task | Source |
|---|---|---|---|---|---|
| Finance | Stock | 3661 | 6 | Generation | (Yoon et al., 2019) |
| | Exchange | 5288 | 8 | Forecasting | (Lai et al., 2018) |
| | SharePriceIncrease | 965 | 1 | Classification | (Middlehurst et al., 2024) |
| Energy | Energy | 19711 | 28 | Generation | (Candanedo et al., 2017) |
| | ETTh1 | 8617 | 7 | Forecasting | (Zhou et al., 2021) |
| Space | MSL | 58294 | 55 | Anomaly Detection | (Hundman et al., 2018) |
| | SMAP | 132458 | 25 | Anomaly Detection | (Hundman et al., 2018) |
| Cloud | PSM | 135160 | 25 | Anomaly Detection | (Abdulaal et al., 2021) |
| | SMD | 7084 | 38 | Anomaly Detection | (Su et al., 2019) |
| ECG | ECG5000 | 200 | 1 | Classification | (Goldberger et al., 2000) |
| | NI-FECG | 120 | 1 | Classification | (Silva et al., 2013) |
| EEG | SelfRegulationSCP2 | 500 | 7 | Classification | (Bagnall et al., 2018) |
| | Blink | 1800 | 4 | Classification | (Chicaiza & Benalcázar, 2021) |
| Sensors | ElectricDevices | 8926 | 1 | Classification | (Lines et al., 2011) |
| | Trace | 100 | 1 | Classification | (Roverso, 2002) |
| | FordB | 3636 | 1 | Classification | (Dau et al., 2019) |
| Human Activity | UWaveGestureLibrary | 2238 | 3 | Classification | (Liu et al., 2009) |
| | EMOPain | 968 | 30 | Classification | (Egede et al., 2020) |
| Traffic | Chinatown | 20 | 1 | Classification | (Dau et al., 2019) |

hyperparameters are summarized in Tables 6, 7, and 8, respectively. Analysis of key hyperparameters are presented in C.2.

### B.4. Evaluation Metrics

Given a limited number of training instances, we require models to generate time series with the same size as the original dataset. This design ensures a fair comparison across different generative models by mitigating potential bias arising from discrepancies in test set size. To maintain consistency with prior works (Gonen et al., 2025; Yao et al., 2025), we adopt widely used evaluation metrics from three complementary perspectives: (1) the distributional proximity between synthetic and real time series, (2) the fidelity of temporal dependencies, and (3) the usefulness in downstream applications. Moreover, preserving both diverse individual-level statistical characteristics and population-level properties inherent to time series can help reduce model bias and further enhance downstream performance in downstream tasks. Accordingly, we further incorporate relevant metrics (Ang et al., 2023; Li et al., 2025) to assess generative performance. Detailed interpretations of these metrics, along with corresponding formulations, are presented below.

**Context-FID score.** It replaces the original image-based FID metric with a time series representations learned by TS2Vec (Yuan & Qiao, 2024). Specifically, both synthetic and real time series are first encoded using a pre-trained TS2Vec model, and the FID score is subsequently computed in the resulting representation space. Lower scores indicate closer alignment between the global and contextual distributions of synthetic and real data, and thus correspond to higher-quality generation.

**Discriminative score.** It is trained on a post-hoc LSTM-based time series classifier (clf) in a supervised manner on the training sets constructed from synthetic and real data, where synthetic samples are labeled as 0 and real samples as 1. The metric is then computed on the corresponding test sets.

$$\left| \frac{\sum_{n=1}^{S}(0 = \text{clf}(x_n^f)) + \sum_{n=1}^{S}(1 = \text{clf}(x_n^r))}{2S} - 0.5 \right|, \quad (16)$$

*Table 5.* List of few-shot evaluation datasets.

| Domain | Dataset | Time Pionts | Variables | Task | Source |
|--------|---------|-------------|-----------|------|--------|
| Physics | MuJoCo | 3000 | 14 | Simulation | (Todorov et al., 2012) |
| Electricity | ETTm1 | 34369 | 7 | Forecasting | (Zhou et al., 2021) |
| | ETTm2 | 34273 | 7 | Forecasting | (Zhou et al., 2021) |
| | ETTh2 | 8353 | 7 | Forecasting | (Zhou et al., 2021) |
| Healthcare | ILI | 581 | 7 | Forecasting | (Wu et al., 2021) |
| Weather | SaugeenRiverFlow | 18921 | 1 | Forecasting | (McLeod & Gweon, 2013) |
| ECG | ECG200 | 200 | 1 | Classification | (Olszewski, 2001) |
| Finance | AAPL | 2518 | 5 | Forecasting | (Gong et al., 2022) |
| Sensor | StarLightCurves | 1000 | 1 | Classification | (Rebbapragada et al., 2009) |
| Nature | AirQuality | 9357 | 13 | Generation | (Yi et al., 2016) |

*Table 6.* Pre-training hyperparameters.

| Parameter | Value |
|-----------|-------|
| Optimizer | AdamW |
| Learning rate | $1 \times 10^{-4}$ |
| Weight decay | $1 \times 10^{-5}$ |
| Batch size | 1024 |
| Epochs | 1000 |
| EMA decay | 0.9999 |

*Table 7.* Fine-tuning hyperparameters.

| Parameter | Value |
|-----------|-------|
| Optimizer | AdamW |
| Learning rate | $[1 \times 10^{-4}, 2 \times 10^{-4}]$ |
| Weight decay | $[1 \times 10^{-5}, 2 \times 10^{-5}]$ |
| Batch size | $\min(2048, \text{subset size})$ |
| Epochs | 1000 |
| EMA decay | 0.9999 |

*Table 8.* Shared model hyperparameters.

| Parameter | Value |
|-----------|-------|
| Diffusion steps | 250 |
| Hidden Dim. | 256 |
| Head Num. | 4 |
| Expert Num. | 8 |
| Activated expert | 2 |
| Layer Num. | 6 |
| Expansion ratio | 4 |

where $S$ is set length, $x_n^r$ is real data, $x_n^f$ is synthetic data.

**Predictive score.** It employs the same LSTM-based neural network as used in the Discriminative Score to evaluate whether synthetic data can effectively support forecasting tasks. Specifically, the LSTM-based model is first trained to predict future values given sequences of generated past time steps. The trained model is then evaluated on the original dataset by computing the Mean Absolute Error (MAE) of the forecasting loss.

**Marginal Distribution Difference (MDD).** This metric constructs empirical histograms for each dimension and time step of the generated series using the bin centers and widths from the original data. The average absolute bin-wise difference between the generated and original histograms across bins is then computed to quantify how closely the distributions align.

**AutoCorrelation Difference (ACD).** It computes the autocorrelation for both the original and generated time series to evaluates the preservation of temporal dependencies.

**Skewness Difference (SD).** This statistical metric quantifies the distributional asymmetry of time series. Specifically, given the means $(\mu^r, \mu^f)$ and standard deviations $(\sigma^r, \sigma^f)$ of the original and generated time series, the fidelity of generation is evaluated by computing the difference in skewness between them as:

$$\text{SD} = \left| \frac{\mathbb{E}[(x^r - \mu^r)^3]}{(\sigma^r)^3} - \frac{\mathbb{E}[(x^f - \mu^f)^3]}{(\sigma^f)^3} \right|. \tag{17}$$

**Kurtosis Difference (KD).** Similar to skewness, kurtosis characterizes the tail behavior of a distribution, capturing the prevalence of extreme deviations from the mean. The kurtosis difference is computed as:

$$\text{KD} = \left| \frac{\mathbb{E}[(x^r - \mu^r)^4]}{(\sigma^r)^4} - \frac{\mathbb{E}[(x^f - \mu^f)^4]}{(\sigma^f)^4} \right|. \tag{18}$$

**Value Distribution Shift (VDS).** It quantifies the population-level distribution shift of generated time series in terms of values:

$$\text{VDS} = \frac{1}{C} \sum_{i=1}^{C} D(P_V^i, Q_V^i), \tag{19}$$

where $D$ denotes a distribution distance measure (e.g., KL divergence), $P_V^i$ represents the value distribution of the $i^{\text{th}}$ channel in real data, and $Q_V^i$ corresponds to the value distribution of the generated data.

**Functional Dependency Distribution Shift (FDDS).** This metric captures population-level distribution shifts in terms of functional dependencies by considering the cross-correlation distribution across all channel pairs:

$$\text{FDDS} = \frac{1}{P} \sum_{p=1}^{P} D(P_{\text{FD}}^{i,j}, Q_{\text{FD}}^{i,j}), \tag{20}$$

where $P$ denotes the total number of channel pairs, $P_{\text{FD}}^{i,j}$ represents the distribution of the functional dependency scores between $i^{\text{th}}$ and $j^{\text{th}}$ channel computed via cross-correlation on the original data, $Q_{\text{FD}}^{i,j}$ denotes the corresponding distribution for the generated data.

## C. Additional Experiment Results

In this section, we present the detailed experiment results omitted from the main body of the paper due to space constraints.

### C.1. Few-shot and Full-shot Generation

Tables 9 to 18 report the Context-FID score, Discriminative score and Predictive score across all datasets. Each table corresponds to a specific dataset with different training subset sizes. The few-shot settings include 5%, 10%, and 15% of the original dataset sampled randomly, as well as extremely low-data scenarios with only 10, 25, and 50 samples are available. The full-shot setting uses the entire dataset for training.

### C.2. Analysis of Key Hyperparameters

To investigate how expert selection affects results, we conduct experiments by varying the number of basis from $\{4, 8, 16, 32\}$ and experts from $\{4, 8, 12, 16\}$, respectively. As shown in Figure 8, increasing the number of basis facilitates more precise target domain inference. However, the performance trend is non-monotonic, as excessive basis may cause overfitting and introduce noise from irrelevant experts. Similarly, in Figure 9, an overly large number of experts results in inferior performance. This can be attributed to the fragmentation of domain knowledge across experts under sparse activation.

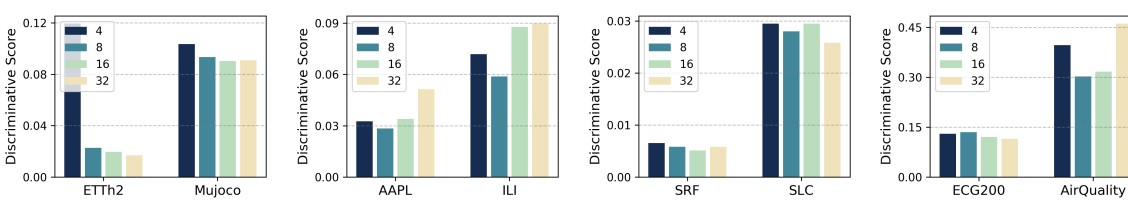

*Figure 8.* Impact of the number of basis within each prototype.

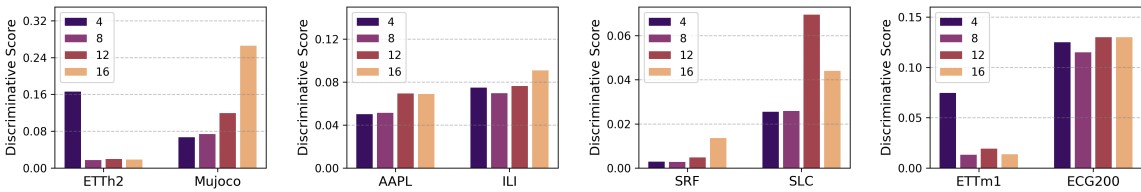

*Figure 9.* Impact of the number of expert within MoDE.

*Table 9.* Main Results - Part 1. Few and full-shot results on AAPL dataset.

| Subset | Metric | TimeMoDE | ImagenFew | ImagenTime | DiffusionTS | KoVAE |
|--------|--------|----------|-----------|------------|-------------|-------|
| 5% | c-FID | **0.303** | 0.419 | 1.830 | 1.996 | 5.841 |
| | Disc. | **0.069±0.025** | 0.085±0.021 | 0.204±0.074 | 0.311±0.009 | 0.357±0.014 |
| | Pred. | 0.514±0.003 | **0.513±0.002** | 0.521±0.003 | 0.518±0.003 | 0.552±0.018 |
| 10% | c-FID | **0.171** | 0.312 | 1.201 | 1.920 | 5.697 |
| | Disc. | **0.051±0.011** | 0.066±0.017 | 0.169±0.132 | 0.303±0.026 | 0.350±0.016 |
| | Pred. | **0.513±0.003** | 0.514±0.003 | 0.520±0.001 | 0.517±0.003 | 0.538±0.010 |
| 15% | c-FID | **0.237** | 0.242 | 1.156 | 2.210 | 2.783 |
| | Disc. | **0.041±0.016** | 0.065±0.007 | 0.168±0.128 | 0.307±0.074 | 0.286±0.016 |
| | Pred. | **0.513±0.004** | 0.514±0.002 | 0.519±0.004 | 0.522±0.005 | 0.521±0.002 |
| 10# | c-FID | **1.953** | 2.585 | 7.927 | 4.676 | 6.188 |
| | Disc. | **0.214±0.056** | 0.236±0.056 | 0.461±0.041 | 0.367±0.061 | 0.399±0.012 |
| | Pred. | **0.519±0.008** | 0.527±0.003 | 0.567±0.010 | 0.568±0.014 | 0.530±0.004 |
| 25# | c-FID | **0.696** | 1.593 | 2.721 | 4.446 | 6.187 |
| | Disc. | **0.087±0.024** | 0.184±0.017 | 0.369±0.086 | 0.350±0.087 | 0.374±0.012 |
| | Pred. | **0.516±0.005** | 0.518±0.003 | 0.523±0.004 | 0.533±0.006 | 0.538±0.010 |
| 50# | c-FID | **0.382** | 0.391 | 1.801 | 1.786 | 6.295 |
| | Disc. | **0.052±0.040** | 0.083±0.015 | 0.242±0.113 | 0.291±0.025 | 0.363±0.010 |
| | Pred. | **0.515±0.002** | 0.516±0.002 | 0.522±0.004 | 0.517±0.003 | 0.563±0.012 |
| Full | c-FID | **0.080** | 0.087 | 0.982 | 1.926 | 0.690 |
| | Disc. | **0.023±0.012** | 0.012±0.006 | 0.174±0.125 | 0.268±0.014 | 0.168±0.042 |
| | Pred. | 0.513±0.001 | 0.514±0.003 | 0.519±0.003 | 0.515±0.002 | **0.512±0.002** |

*Table 10.* Main Results - Part 2. Few and full-shot results on AirQuality dataset.

| Subset | Metric | TimeMoDE | ImagenFew | ImagenTime | DiffusionTS | KoVAE |
|--------|--------|----------|-----------|------------|-------------|-------|
| 5% | c-FID | **0.865** | 0.955 | 5.349 | 2.167 | 2.095 |
| | Disc. | 0.412±0.027 | 0.404±0.028 | 0.490±0.004 | 0.409±0.031 | **0.327±0.036** |
| | Pred. | **0.007±0.001** | 0.010±0.001 | 0.017±0.004 | 0.049±0.020 | 0.048±0.002 |
| 10% | c-FID | **0.845** | 1.066 | 5.135 | 2.319 | 1.505 |
| | Disc. | 0.365±0.017 | 0.366±0.022 | 0.489±0.006 | 0.374±0.077 | **0.259±0.023** |
| | Pred. | **0.008±0.001** | 0.009±0.001 | 0.018±0.008 | 0.031±0.028 | 0.041±0.000 |
| 15% | c-FID | **0.697** | 1.408 | 5.085 | 2.62 | 1.311 |
| | Disc. | 0.353±0.018 | 0.325±0.040 | 0.490±0.004 | 0.431±0.027 | **0.284±0.060** |
| | Pred. | **0.008±0.001** | 0.009±0.000 | 0.016±0.002 | 0.012±0.001 | 0.034±0.008 |
| 10# | c-FID | **1.719** | 2.386 | 9.336 | 4.918 | 3.943 |
| | Disc. | **0.325±0.043** | 0.498±0.002 | 0.499±0.001 | 0.341±0.121 | 0.457±0.018 |
| | Pred. | **0.041±0.000** | 0.041±0.002 | 0.044±0.000 | 0.045±0.000 | 0.043±0.000 |
| 25# | c-FID | **0.716** | 1.347 | 7.543 | 3.199 | 3.146 |
| | Disc. | **0.328±0.029** | 0.496±0.005 | 0.498±0.002 | 0.447±0.032 | 0.389±0.035 |
| | Pred. | **0.007±0.001** | 0.025±0.003 | 0.044±0.000 | 0.041±0.000 | 0.043±0.000 |
| 50# | c-FID | **0.468** | 0.889 | 6.554 | 3.72 | 2.877 |
| | Disc. | **0.273±0.021** | 0.492±0.006 | 0.497±0.002 | 0.456±0.015 | 0.372±0.036 |
| | Pred. | **0.013±0.002** | 0.019±0.002 | 0.044±0.000 | 0.038±0.006 | 0.040±0.001 |
| Full | c-FID | **0.032** | 0.094 | 0.266 | 2.437 | 1.063 |
| | Disc. | 0.154±0.007 | **0.051±0.006** | 0.064±0.005 | 0.335±0.124 | 0.237±0.018 |
| | Pred. | **0.006±0.000** | **0.006±0.000** | 0.008±0.000 | 0.033±0.035 | 0.016±0.012 |

*Table 11.* Main Results - Part 3. Few and full-shot results on ECG200 dataset.

| Subset | Metric | TimeMoDE | ImagenFew | ImagenTime | DiffusionTS | KoVAE |
|--------|--------|----------|-----------|------------|-------------|-------|
| 5% | c-FID | 0.793 | **0.784** | 0.812 | 1.803 | 1.130 |
|  | Disc. | **0.105±0.049** | 0.172±0.075 | 0.180±0.058 | 0.360±0.066 | 0.107±0.084 |
|  | Pred. | **1.061±0.000** | 1.062±0.000 | **1.061±0.000** | 1.065±0.000 | 1.062±0.000 |
| 10% | c-FID | **0.535** | 0.550 | 0.760 | 2.233 | 1.207 |
|  | Disc. | **0.115±0.087** | 0.135±0.082 | 0.130±0.061 | 0.355±0.069 | 0.182±0.128 |
|  | Pred. | **1.061±0.000** | 1.062±0.000 | **1.061±0.000** | 1.062±0.000 | 1.062±0.000 |
| 15% | c-FID | 0.304 | **0.254** | 0.402 | 2.308 | 0.788 |
|  | Disc. | 0.040±0.042 | **0.022±0.033** | 0.085±0.058 | 0.375±0.045 | 0.075±0.052 |
|  | Pred. | **1.061±0.000** | 1.062±0.000 | 1.062±0.000 | 1.064±0.000 | 1.062±0.000 |
| 10# | c-FID | 0.478 | **0.385** | 0.704 | 2.588 | 1.207 |
|  | Disc. | 0.133±0.086 | **0.107±0.060** | 0.160±0.023 | 0.375±0.061 | 0.188±0.094 |
|  | Pred. | **1.061±0.000** | 1.062±0.000 | **1.061±0.000** | **1.061±0.000** | 1.062±0.000 |
| 25# | c-FID | **0.198** | 0.264 | 0.353 | 2.438 | 1.053 |
|  | Disc. | 0.090±0.067 | **0.065±0.041** | 0.135±0.046 | 0.365±0.051 | 0.107±0.055 |
|  | Pred. | **1.062±0.000** | **1.062±0.000** | **1.062±0.000** | 1.063±0.000 | 1.063±0.000 |
| 50# | c-FID | **0.103** | 0.156 | 0.168 | 2.501 | 0.876 |
|  | Disc. | 0.060±0.044 | **0.050±0.032** | **0.050±0.054** | 0.383±0.050 | 0.140±0.073 |
|  | Pred. | **1.062±0.000** | **1.062±0.000** | **1.062±0.000** | 1.063±0.000 | **1.062±0.000** |
| Full | c-FID | 0.072 | 0.118 | 0.173 | 2.027 | **0.610** |
|  | Disc. | **0.035±0.030** | 0.115±0.085 | 0.075±0.039 | 0.345±0.070 | 0.070±0.037 |
|  | Pred. | **1.062±0.000** | 1.063±0.000 | 1.063±0.000 | **1.062±0.000** | **1.062±0.000** |

*Table 12.* Main Results - Part 4. Few and full-shot results on ETTh2 dataset.

| Subset | Metric | TimeMoDE | ImagenFew | ImagenTime | DiffusionTS | KoVAE |
|--------|--------|----------|-----------|------------|-------------|-------|
| 5% | c-FID | **0.131** | 0.211 | 0.607 | 2.509 | 4.036 |
|  | Disc. | **0.037±0.010** | 0.058±0.025 | 0.261±0.104 | 0.315±0.024 | 0.445±0.012 |
|  | Pred. | **0.682±0.004** | 0.688±0.002 | 0.704±0.002 | 0.734±0.006 | 0.784±0.005 |
| 10% | c-FID | **0.074** | 0.250 | 0.517 | 1.994 | 2.966 |
|  | Disc. | **0.017±0.008** | 0.043±0.011 | 0.284±0.115 | 0.255±0.024 | 0.407±0.017 |
|  | Pred. | **0.677±0.006** | 0.685±0.002 | 0.701±0.002 | 0.761±0.025 | 0.736±0.011 |
| 15% | c-FID | **0.058** | 0.103 | 0.53 | 2.325 | 2.433 |
|  | Disc. | **0.022±0.038** | 0.024±0.007 | 0.223±0.116 | 0.316±0.031 | 0.386±0.022 |
|  | Pred. | **0.676±0.004** | 0.681±0.005 | 0.699±0.002 | 0.772±0.024 | 0.739±0.004 |
| 10# | c-FID | 2.673 | **2.478** | 3.031 | 4.556 | 7.149 |
|  | Disc. | 0.379±0.058 | **0.355±0.074** | 0.457±0.030 | 0.480±0.014 | 0.488±0.004 |
|  | Pred. | 0.745±0.010 | **0.742±0.013** | 0.738±0.002 | 0.819±0.005 | 0.810±0.003 |
| 25# | c-FID | **1.704** | 1.895 | 2.279 | 2.755 | 6.044 |
|  | Disc. | **0.268±0.052** | 0.285±0.029 | 0.401±0.069 | 0.432±0.038 | 0.480±0.004 |
|  | Pred. | 0.719±0.017 | 0.721±0.016 | **0.714±0.002** | 0.816±0.004 | 0.812±0.003 |
| 50# | c-FID | **1.023** | 1.381 | 1.960 | 2.285 | 5.346 |
|  | Disc. | **0.204±0.021** | 0.214±0.019 | 0.391±0.091 | 0.347±0.018 | 0.484±0.006 |
|  | Pred. | 0.712±0.007 | **0.711±0.003** | 0.716±0.005 | 0.772±0.023 | 0.808±0.006 |
| Full | c-FID | 0.006 | **0.053** | 0.139 | 1.770 | 1.105 |
|  | Disc. | 0.012±0.006 | **0.011±0.006** | 0.018±0.007 | 0.255±0.024 | 0.230±0.018 |
|  | Pred. | **0.671±0.004** | 0.675±0.003 | 0.681±0.001 | 0.724±0.011 | 0.704±0.002 |

*Table 13.* Main Results - Part 5. Few and full-shot results on ETTm1 dataset.

| Subset | Metric | TimeMoDE | ImagenFew | ImagenTime | DiffusionTS | KoVAE |
|--------|--------|----------|-----------|------------|-------------|-------|
| 5% | c-FID | **0.039** | 0.129 | 1.823 | 1.734 | 3.106 |
|    | Disc. | **0.020±0.007** | 0.042±0.010 | 0.400±0.086 | 0.322±0.025 | 0.415±0.016 |
|    | Pred. | **0.677±0.005** | 0.682±0.003 | 0.698±0.003 | 0.777±0.027 | 0.714±0.003 |
| 10% | c-FID | **0.030** | 0.052 | 0.293 | 1.793 | 2.068 |
|     | Disc. | **0.013±0.006** | 0.023±0.005 | 0.068±0.013 | 0.323±0.021 | 0.360±0.017 |
|     | Pred. | **0.679±0.004** | 0.680±0.002 | 0.686±0.002 | 0.853±0.011 | 0.712±0.005 |
| 15% | c-FID | **0.020** | 0.034 | 0.115 | 1.975 | 1.571 |
|     | Disc. | **0.009±0.004** | 0.018±0.007 | 0.046±0.009 | 0.319±0.011 | 0.288±0.026 |
|     | Pred. | 0.678±0.003 | **0.676±0.003** | 0.681±0.004 | 0.756±0.006 | 0.711±0.002 |
| 10# | c-FID | **3.678** | 3.923 | 6.292 | 4.058 | 6.512 |
|     | Disc. | 0.434±0.031 | **0.423±0.033** | 0.452±0.029 | 0.451±0.017 | 0.480±0.009 |
|     | Pred. | 0.825±0.025 | 0.836±0.015 | **0.813±0.003** | 0.899±0.009 | 0.821±0.012 |
| 25# | c-FID | **1.831** | 2.110 | 7.454 | 2.929 | 6.603 |
|     | Disc. | **0.244±0.037** | 0.368±0.080 | 0.484±0.031 | 0.401±0.016 | 0.488±0.002 |
|     | Pred. | 0.753±0.021 | 0.762±0.007 | **0.742±0.003** | 0.847±0.013 | 0.778±0.012 |
| 50# | c-FID | **1.003** | 1.308 | 5.131 | 2.544 | 5.78 |
|     | Disc. | **0.194±0.029** | 0.322±0.098 | 0.481±0.019 | 0.366±0.016 | 0.485±0.007 |
|     | Pred. | 0.708±0.011 | 0.705±0.005 | **0.695±0.002** | 0.864±0.030 | 0.745±0.011 |
| Full | c-FID | **0.006** | 0.009 | 0.014 | 1.913 | 0.627 |
|      | Disc. | 0.005±0.003 | **0.003±0.003** | 0.007±0.003 | 0.314±0.018 | 0.172±0.017 |
|      | Pred. | **0.672±0.001** | 0.676±0.004 | 0.674±0.003 | 0.758±0.012 | 0.711±0.005 |

*Table 14.* Main Results - Part 6. Few and full-shot results on ETTm2 dataset.

| Subset | Metric | TimeMoDE | ImagenFew | ImagenTime | DiffusionTS | KoVAE |
|--------|--------|----------|-----------|------------|-------------|-------|
| 5% | c-FID | **0.044** | 0.095 | 0.769 | 1.663 | 2.824 |
|    | Disc. | **0.020±0.010** | 0.025±0.011 | 0.346±0.112 | 0.227±0.026 | 0.374±0.027 |
|    | Pred. | **0.694±0.003** | 0.699±0.002 | 0.725±0.002 | 0.803±0.024 | 0.750±0.004 |
| 10% | c-FID | **0.024** | 0.060 | 0.319 | 1.661 | 1.274 |
|     | Disc. | **0.012±0.007** | 0.016±0.004 | 0.066±0.031 | 0.242±0.020 | 0.235±0.029 |
|     | Pred. | **0.694±0.002** | 0.697±0.003 | 0.709±0.002 | 0.774±0.026 | 0.727±0.003 |
| 15% | c-FID | **0.018** | 0.077 | 0.222 | 1.705 | 0.817 |
|     | Disc. | **0.012±0.005** | 0.023±0.005 | 0.048±0.008 | 0.244±0.014 | 0.149±0.026 |
|     | Pred. | **0.696±0.004** | 0.696±0.002 | 0.706±0.002 | 0.763±0.014 | 0.727±0.004 |
| 10# | c-FID | **3.361** | 3.790 | 4.467 | 5.86 | 8.665 |
|     | Disc. | 0.381±0.049 | 0.384±0.030 | **0.370±0.023** | 0.438±0.007 | 0.473±0.006 |
|     | Pred. | 0.804±0.004 | 0.805±0.003 | **0.784±0.024** | 0.969±0.063 | 0.804±0.020 |
| 25# | c-FID | 1.824 | **1.813** | 3.385 | 2.978 | 6.302 |
|     | Disc. | **0.248±0.036** | 0.334±0.092 | 0.486±0.004 | 0.369±0.018 | 0.452±0.008 |
|     | Pred. | **0.747±0.017** | 0.754±0.014 | 0.766±0.003 | 0.888±0.027 | 0.811±0.002 |
| 50# | c-FID | **0.920** | 1.095 | 2.111 | 2.633 | 6.031 |
|     | Disc. | **0.166±0.016** | 0.371±0.094 | 0.470±0.006 | 0.328±0.031 | 0.453±0.010 |
|     | Pred. | 0.726±0.006 | **0.725±0.004** | 0.743±0.003 | 0.787±0.006 | 0.810±0.003 |
| Full | c-FID | **0.005** | 0.024 | 0.031 | 1.706 | 0.571 |
|      | Disc. | 0.007±0.005 | 0.011±0.003 | **0.006±0.005** | 0.205±0.017 | 0.124±0.019 |
|      | Pred. | 0.693±0.003 | **0.692±0.002** | 0.694±0.002 | 0.732±0.006 | 0.728±0.004 |

*Table 15.* Main Results - Part 7. Few and full-shot results on ILI dataset.

| Subset | Metric | TimeMoDE | ImagenFew | ImagenTime | DiffusionTS | KoVAE |
|---|---|---|---|---|---|---|
| 5% | c-FID | **1.042** | 1.110 | 6.663 | 6.256 | 6.105 |
|  | Disc. | **0.177±0.036** | 0.359±0.101 | 0.483±0.020 | 0.440±0.018 | 0.493±0.005 |
|  | Pred. | **0.549±0.003** | 0.564±0.004 | 0.579±0.003 | 0.617±0.021 | 0.644±0.011 |
| 10% | c-FID | **0.450** | 0.617 | 3.731 | 4.158 | 7.691 |
|  | Disc. | **0.089±0.030** | 0.223±0.115 | 0.470±0.039 | 0.399±0.031 | 0.490±0.006 |
|  | Pred. | **0.546±0.003** | 0.561±0.004 | 0.575±0.003 | 0.581±0.006 | 0.664±0.012 |
| 15% | c-FID | **0.261** | 0.439 | 2.77 | 3.864 | 4.509 |
|  | Disc. | **0.030±0.016** | 0.078±0.058 | 0.466±0.036 | 0.428±0.027 | 0.490±0.005 |
|  | Pred. | **0.542±0.002** | 0.557±0.003 | 0.574±0.003 | 0.572±0.003 | 0.615±0.012 |
| 10# | c-FID | 2.244 | **2.127** | 7.736 | 5.487 | 6.538 |
|  | Disc. | **0.307±0.071** | 0.351±0.069 | 0.484±0.017 | 0.466±0.039 | 0.479±0.016 |
|  | Pred. | **0.592±0.015** | 0.595±0.011 | 0.612±0.016 | 0.663±0.046 | 0.647±0.014 |
| 25# | c-FID | 1.202 | **1.130** | 5.846 | 5.891 | 6.346 |
|  | Disc. | **0.217±0.036** | 0.282±0.086 | 0.482±0.018 | 0.455±0.028 | 0.490±0.007 |
|  | Pred. | **0.557±0.011** | 0.568±0.007 | 0.589±0.006 | 0.594±0.005 | 0.649±0.027 |
| 50# | c-FID | **0.653** | 0.718 | 4.486 | 4.058 | 6.317 |
|  | Disc. | **0.114±0.028** | 0.235±0.129 | 0.468±0.039 | 0.417±0.037 | 0.488±0.008 |
|  | Pred. | **0.552±0.005** | 0.561±0.003 | 0.574±0.003 | 0.577±0.004 | 0.780±0.027 |
| Full | c-FID | **0.115** | 1.577 | 2.042 | 2.525 | 3.629 |
|  | Disc. | **0.047±0.053** | 0.395±0.066 | 0.403±0.119 | 0.388±0.022 | 0.453±0.024 |
|  | Pred. | **0.543±0.001** | 0.562±0.001 | 0.572±0.002 | 0.566±0.007 | 0.572±0.008 |

*Table 16.* Main Results - Part 8. Few and full-shot results on Mujoco dataset.

| Subset | Metric | TimeMoDE | ImagenFew | ImagenTime | DiffusionTS | KoVAE |
|---|---|---|---|---|---|---|
| 5% | c-FID | **0.197** | 0.417 | 8.968 | 10.259 | 1.538 |
|  | Disc. | **0.120±0.017** | 0.489±0.010 | 0.499±0.001 | 0.495±0.004 | 0.358±0.029 |
|  | Pred. | **0.039±0.002** | 0.040±0.001 | 0.063±0.001 | 0.085±0.012 | 0.050±0.002 |
| 10% | c-FID | **0.161** | 0.319 | 8.747 | 10.076 | 0.914 |
|  | Disc. | **0.091±0.017** | 0.343±0.130 | 0.497±0.004 | 0.494±0.005 | 0.264±0.031 |
|  | Pred. | **0.035±0.001** | 0.040±0.002 | 0.063±0.002 | 0.080±0.008 | 0.046±0.002 |
| 15% | c-FID | **0.193** | 0.246 | 8.499 | 0.703 | 0.594 |
|  | Disc. | **0.100±0.012** | 0.202±0.086 | 0.499±0.001 | 0.361±0.082 | 0.221±0.029 |
|  | Pred. | **0.035±0.002** | 0.040±0.001 | 0.062±0.003 | 0.048±0.001 | 0.037±0.002 |
| 10# | c-FID | **2.321** | 3.679 | 13.111 | 11.809 | 5.182 |
|  | Disc. | **0.306±0.079** | 0.499±0.001 | 0.498±0.004 | 0.449±0.017 | 0.456±0.024 |
|  | Pred. | 0.118±0.023 | **0.079±0.004** | 0.092±0.002 | 0.184±0.004 | 0.094±0.004 |
| 25# | c-FID | **1.418** | 2.546 | 11.395 | 4.244 | 5.175 |
|  | Disc. | **0.285±0.064** | 0.499±0.002 | 0.499±0.001 | 0.386±0.039 | 0.452±0.016 |
|  | Pred. | 0.066±0.011 | **0.058±0.003** | 0.062±0.001 | 0.061±0.002 | 0.072±0.004 |
| 50# | c-FID | **0.578** | 1.398 | 10.461 | 1.893 | 4.752 |
|  | Disc. | **0.195±0.001** | 0.499±0.001 | 0.499±0.001 | 0.345±0.017 | 0.458±0.015 |
|  | Pred. | 0.052±0.002 | **0.050±0.002** | 0.066±0.002 | 0.060±0.001 | 0.080±0.004 |
| Full | c-FID | **0.031** | 0.094 | 0.938 | 0.879 | 0.311 |
|  | Disc. | **0.035±0.017** | 0.046±0.011 | 0.300±0.124 | 0.223±0.012 | 0.149±0.022 |
|  | Pred. | **0.033±0.003** | 0.034±0.001 | 0.041±0.001 | 0.042±0.000 | 0.039±0.002 |

*Table 17.* Main Results - Part 9. Few and full-shot results on SaugeenRiverFlow (SRF) dataset.

| Subset | Metric | TimeMoDE | ImagenFew | ImagenTime | DiffusionTS | KoVAE |
|--------|--------|----------|-----------|------------|-------------|-------|
| 5% | c-FID | 0.070 | **0.023** | 0.052 | 1.272 | 1.712 |
| | Disc. | 0.010±0.007 | 0.012±0.004 | **0.007±0.004** | 0.175±0.042 | 0.149±0.058 |
| | Pred. | 0.604±0.000 | 0.604±0.000 | 0.604±0.000 | **0.603±0.000** | 0.606±0.000 |
| 10% | c-FID | 0.053 | **0.026** | 0.043 | 1.249 | 0.463 |
| | Disc. | 0.005±0.005 | 0.012±0.005 | **0.004±0.003** | 0.168±0.028 | 0.074±0.029 |
| | Pred. | 0.604±0.000 | 0.604±0.000 | 0.604±0.000 | **0.603±0.000** | 0.604±0.000 |
| 15% | c-FID | 0.030 | 0.021 | 0.016 | 1.403 | 0.269 |
| | Disc. | 0.009±0.003 | 0.017±0.004 | **0.005±0.005** | 0.204±0.007 | 0.027±0.013 |
| | Pred. | 0.604±0.000 | 0.604±0.000 | 0.604±0.000 | **0.602±0.000** | 0.605±0.000 |
| 10# | c-FID | 0.857 | 1.331 | **0.672** | 3.368 | 2.21 |
| | Disc. | 0.110±0.027 | 0.150±0.032 | **0.090±0.017** | 0.313±0.012 | 0.218±0.030 |
| | Pred. | **0.603±0.000** | 0.604±0.000 | 0.604±0.000 | 0.604±0.000 | **0.603±0.000** |
| 25# | c-FID | 0.658 | 0.911 | **0.302** | 3.855 | 2.098 |
| | Disc. | 0.044±0.036 | 0.058±0.040 | **0.023±0.017** | 0.350±0.006 | 0.189±0.047 |
| | Pred. | 0.604±0.000 | 0.604±0.000 | 0.604±0.000 | **0.603±0.000** | 0.604±0.000 |
| 50# | c-FID | 0.355 | 0.382 | **0.240** | 2.038 | 1.766 |
| | Disc. | **0.026±0.021** | 0.049±0.029 | **0.026±0.011** | 0.256±0.005 | 0.158±0.059 |
| | Pred. | 0.604±0.000 | 0.604±0.000 | 0.604±0.000 | **0.602±0.000** | 0.603±0.000 |
| Full | c-FID | 0.029 | 0.011 | **0.009** | 1.679 | 0.292 |
| | Disc. | 0.004±0.002 | 0.006±0.004 | **0.003±0.002** | 0.231±0.007 | 0.027±0.015 |
| | Pred. | 0.605±0.000 | 0.605±0.000 | 0.605±0.000 | **0.602±0.000** | 0.605±0.000 |

*Table 18.* Main Results - Part 10. Few and full-shot results on StarLightCurves (SLC) dataset.

| Subset | Metric | TimeMoDE | ImagenFew | ImagenTime | DiffusionTS | KoVAE |
|--------|--------|----------|-----------|------------|-------------|-------|
| 5% | c-FID | 0.064 | 0.114 | **0.041** | 0.381 | 2.373 |
| | Disc. | 0.027±0.019 | 0.042±0.021 | **0.024±0.016** | 0.168±0.050 | 0.189±0.035 |
| | Pred. | 0.501±0.000 | 0.512±0.000 | 0.500±0.000 | 0.589±0.001 | **0.498±0.000** |
| 10% | c-FID | **0.056** | 0.085 | 0.093 | 0.152 | 2.343 |
| | Disc. | **0.024±0.018** | 0.029±0.023 | 0.041±0.022 | 0.063±0.035 | 0.208±0.051 |
| | Pred. | **0.499±0.000** | 0.500±0.000 | **0.499±0.000** | 0.535±0.001 | **0.499±0.000** |
| 15% | c-FID | **0.019** | 0.024 | 0.053 | 0.109 | 0.274 |
| | Disc. | 0.020±0.013 | **0.013±0.008** | 0.027±0.017 | 0.025±0.018 | 0.077±0.032 |
| | Pred. | **0.499±0.000** | 0.500±0.000 | 0.500±0.000 | 0.521±0.001 | 0.510±0.001 |
| 10# | c-FID | **0.211** | 0.308 | 0.275 | 0.496 | 2.663 |
| | Disc. | 0.048±0.050 | **0.032±0.021** | 0.051±0.048 | 0.085±0.054 | 0.280±0.106 |
| | Pred. | 0.515±0.000 | **0.500±0.000** | 0.519±0.001 | 0.530±0.001 | 0.553±0.000 |
| 25# | c-FID | 0.045 | 0.320 | **0.033** | 0.874 | 2.113 |
| | Disc. | 0.017±0.014 | 0.098±0.037 | **0.015±0.011** | 0.208±0.129 | 0.248±0.070 |
| | Pred. | 0.508±0.000 | 0.527±0.001 | **0.497±0.000** | 0.632±0.001 | 0.513±0.000 |
| 50# | c-FID | 0.074 | 0.187 | **0.042** | 0.381 | 2.373 |
| | Disc. | 0.039±0.018 | 0.085±0.021 | **0.031±0.020** | 0.154±0.056 | 0.182±0.039 |
| | Pred. | 0.507±0.000 | 0.520±0.001 | 0.500±0.000 | 0.589±0.001 | **0.498±0.000** |
| Full | c-FID | 0.023 | 0.017 | **0.005** | 0.098 | 0.103 |
| | Disc. | 0.018±0.011 | 0.022±0.013 | **0.011±0.004** | 0.031±0.010 | 0.033±0.010 |
| | Pred. | **0.497±0.000** | **0.497±0.000** | 0.498±0.000 | 0.513±0.001 | 0.498±0.000 |

## C.3. Ablation Study

Tables 19 report the performance of TimeMoDE compared with designed variants. Our model achieves performance gains from the proposed components in most cases. The improvement is particularly pronounced in the w/ Dense Layer and From Scratch variants, demonstrating the benefit of leveraging a unified architecture to acquire transferable knowledge and advanced MoE designs. Our exploration provides valuable guidance for future research on time series generation.

*Table 19.* Ablation study under 10% of dataset.

| Dataset | Metric | TimeMoDE | w/o MoDE Router | w/ Dense Layer | From Scratch | Timestep Uncond. | Prototype Finetuning |
|---|---|---|---|---|---|---|---|
| AAPL | c-FID | 0.171 | 0.281 | 0.417 | 0.398 | 0.222 | 0.205 |
| | Disc. | 0.051±0.011 | 0.043±0.026 | 0.131±0.044 | 0.085±0.033 | 0.056±0.021 | 0.035±0.013 |
| | Pred. | 0.513±0.003 | 0.513±0.002 | 0.519±0.004 | 0.515±0.002 | 0.515±0.001 | 0.515±0.002 |
| AirQuality | c-FID | 0.845 | 0.580 | 2.647 | 1.350 | 0.701 | 0.709 |
| | Disc. | 0.365±0.017 | 0.371±0.020 | 0.398±0.154 | 0.255±0.023 | 0.350±0.023 | 0.457±0.017 |
| | Pred. | 0.008±0.01 | 0.007±0.000 | 0.016±0.002 | 0.006±0.000 | 0.010±0.001 | 0.007±0.000 |
| ECG200 | c-FID | 0.535 | 0.666 | 1.335 | 2.665 | 0.453 | 0.493 |
| | Disc. | 0.115±0.087 | 0.135±0.111 | 0.260±0.154 | 0.280±0.043 | 0.130±0.051 | 0.115±0.051 |
| | Pred. | 1.061±0.000 | 1.061±0.000 | 1.062±0.000 | 1.061±0.000 | 1.061±0.000 | 1.061±0.000 |
| ETTh2 | c-FID | 0.074 | 0.155 | 1.762 | 0.641 | 0.106 | 0.094 |
| | Disc. | 0.017±0.008 | 0.060±0.028 | 0.384±0.019 | 0.102±0.067 | 0.062±0.053 | 0.028±0.005 |
| | Pred. | 0.677±0.006 | 0.679±0.002 | 0.883±0.015 | 0.686±0.006 | 0.680±0.001 | 0.677±0.003 |
| ETTm1 | c-FID | 0.030 | 0.065 | 0.089 | 0.331 | 0.039 | 0.034 |
| | Disc. | 0.013±0.006 | 0.019±0.004 | 0.044±0.008 | 0.055±0.014 | 0.018±0.002 | 0.012±0.004 |
| | Pred. | 0.679±0.004 | 0.681±0.003 | 0.682±0.002 | 0.684±0.001 | 0.677±0.001 | 0.679±0.002 |
| ETTm2 | c-FID | 0.024 | 0.075 | 0.110 | 0.437 | 0.040 | 0.042 |
| | Disc. | 0.012±0.007 | 0.070±0.045 | 0.288±0.058 | 0.106±0.013 | 0.035±0.014 | 0.019±0.006 |
| | Pred. | 0.694±0.002 | 0.699±0.002 | 0.711±0.006 | 0.700±0.001 | 0.696±0.005 | 0.692±0.002 |
| ILI | c-FID | 0.450 | 0.539 | 0.593 | 1.841 | 0.452 | 0.470 |
| | Disc. | 0.089±0.030 | 0.116±0.043 | 0.172±0.076 | 0.450±0.021 | 0.105±0.029 | 0.096±0.011 |
| | Pred. | 0.546±0.003 | 0.552±0.003 | 0.551±0.003 | 0.576±0.002 | 0.545±0.001 | 0.546±0.003 |
| Mujoco | c-FID | 0.161 | 0.480 | 1.515 | 0.352 | 0.180 | 0.221 |
| | Disc. | 0.091±0.017 | 0.171±0.032 | 0.391±0.020 | 0.185±0.049 | 0.189±0.012 | 0.092±0.015 |
| | Pred. | 0.035±0.001 | 0.041±0.001 | 0.076±0.006 | 0.041±0.001 | 0.038±0.001 | 0.035±0.002 |
| SRF | c-FID | 0.053 | 0.054 | 0.169 | 0.727 | 0.133 | 0.066 |
| | Disc. | 0.005±0.005 | 0.008±0.006 | 0.011±0.003 | 0.102±0.011 | 0.005±0.003 | 0.005±0.002 |
| | Pred. | 0.604±0.000 | 0.604±0.000 | 0.604±0.000 | 0.603±0.000 | 0.604±0.000 | 0.604±0.000 |
| SLC | c-FID | 0.056 | 0.127 | 0.114 | 0.081 | 0.083 | 0.085 |
| | Disc. | 0.024±0.018 | 0.028±0.017 | 0.030±0.020 | 0.048±0.022 | 0.027±0.016 | 0.045±0.014 |
| | Pred. | 0.499±0.000 | 0.497±0.000 | 0.497±0.000 | 0.497±0.000 | 0.497±0.000 | 0.498±0.000 |

