# OpenReview forum: "Towards a Unified Generative Model for Scarce Time Series with Domain Experts"
_ICML.cc/2026/Conference — ICML 2026 regular_

### Official Review · Reviewer_b8gK · 2026-03-04

**Soundness:** 3
**Presentation:** 3
**Significance:** 3
**Originality:** 3
**Overall Recommendation:** 4
**Confidence:** 5

**Summary:**

This paper proposes a unified framework for few-shot generative modeling that leverages a set of learned prototypes to represent the structure of the data. The method learns a prototype-based latent representation that can capture common patterns across samples and uses these prototypes to guide generation in low-data regimes. The framework is designed to work across multiple generative settings and aims to improve generalization when only a small number of training examples are available. The proposed architecture combines prototype learning with a generative model, allowing the model to capture both shared structure and sample-specific variability. Experiments are conducted on several benchmarks, and the results suggest that the approach performs competitively compared to existing few-shot generative models.

**Compliance With Llm Reviewing Policy:**

Affirmed.

**Key Questions For Authors:**

see above

**Limitations:**

Yes

**Strengths And Weaknesses:**

Strengths

Soundness: The paper addresses an important problem in generative modeling, namely how to perform generation when only a few examples are available. The idea of learning a small set of prototypes to capture shared structure is intuitive and well motivated. The overall architecture is described in reasonable detail and the proposed approach appears technically sound. The experimental section evaluates the model on several benchmarks and compares it with existing methods in the literature.

Significance: Few-shot generative modeling is a challenging and relevant problem with applications in many areas such as image synthesis, data augmentation, and scientific data generation. A unified framework that can handle this setting in a principled way could be useful for future work. The prototype-based formulation may also inspire related approaches for other types of generative models.

Originality: The use of learned prototypes as a structural representation in few-shot generative modeling is an interesting idea. While the framework builds on existing generative modeling techniques, the combination with prototype learning provides a new perspective on how to handle low-data regimes.



Weaknesses

Soundness: While the architecture is described in relative detail, some parts remain unclear. For example, it is not entirely clear how the model handles time series with a different number of variates. Since many real-world datasets have varying dimensionality, clarifying how the framework accommodates this would strengthen the paper. Another aspect that would benefit from further analysis is the role of the number of prototypes K. It would be useful to understand how sensitive the performance is to this parameter and how it affects the trade-off between model capacity and generalization.

Presentation: Although the overall structure of the paper is clear, the text could be improved in terms of English writing. Some sentences are slightly hard to follow, and a careful proofreading would improve readability.

Significance: The experimental results are promising, but some clarifications would make the evaluation stronger. In particular, the results reported for the baselines in Tables 1 and 2 appear to differ from the numbers reported in the ImagenFew paper, even though the same benchmark seems to be used. It would be helpful if the authors could clarify the reason for this discrepancy.

---

> ### Author Rebuttal · Authors · 2026-03-30
>
> We thank Reviewer b8gk for recognizing the novelty and contributions of our work. Below, we try to address each point in detail to clarify aspects that were not fully explained and further strengthen our work.
>
> **W1** *While the architecture is described in relative detail, some parts remain unclear. For example, it is not entirely clear how the model handles time series with a different number of variates.*
>
> Due to space and page limitations, we omit some data processing details that are not central to the primary contributions of the manuscript. We appreciate reviewer's point about the clarity of this aspect, and we will include the following additional explanation in the appendix to provide a more comprehensive understanding of our model structure and methodology.
>
> To handle time series with varying numbers of variates, we employ adaptive padding and a dynamic binary mask at runtime. The mask indicates which channels are meaningful and which are padded, allowing the model to focus on the real existing variables while ignoring the padding introduced for uniform formulation of input. This design allows TimeMoDE to train on datasets with varying dimensionalities and learn diverse multivariate correlations. It also improves generalization to downstream tasks involving datasets with uncertain numbers of variates.
>
> **W2** *Another aspect that would benefit from further analysis is the role of the number of prototypes K. It would be useful to understand how sensitive the performance is to this parameter and how it affects the trade-off between model capacity and generalization.*
>
> We vary the number of prototypes $K$ from {$4, 8, 12, 16$} in Figure 9 in the appendix. As presented in Table below, increasing $K$ within a certain range facilitates more precise target domain adaptation, as each prototype focused on fewer patterns can capture more specific and fine-grained temporal features for better inference. However, the performance trend is non-monotonic. An excessively large $K$ can lead to overfitting, as the prototypes may overly focused on fragmented temporal features or even noise, neglecting critical characteristic emerging from the interactions between individual features. Based on our experiments, we set $K=8$ in TimeMoDE.
>
> |  $K$ | ECG200 | ETTh2 | ETTm1 |  ILI  | Mujoco |  SLC  |
> |:--:|:------:|:-----:|:-----:|:-----:|:------:|:-----:|
> |  4 |  0.125 | 0.166 | 0.075 | 0.075 |  0.067 | 0.026 |
> |  8 |  0.116 | 0.017 | 0.013 | 0.069 |  0.074 | 0.025 |
> | 12 |  0.130 | 0.021 | 0.019 | 0.076 |  0.120 | 0.070 |
> | 16 |  0.132 | 0.019 | 0.016 | 0.091 |  0.266 | 0.044 |
>
> **W3** *The experimental results are promising, but some clarifications would make the evaluation stronger. In particular, the results reported for the baselines in Tables 1 and 2 appear to differ from the numbers reported in the ImagenFew paper, even though the same benchmark seems to be used. It would be helpful if the authors could clarify the reason for this discrepancy.*
>
> We appreciate Reviewer for the thoughtful question and would like to clarify the discrepancy in the reported results.
> The results reported in Tables 1 and 2 are averages over ten fine-tuned datasets, which slightly differ from the datasets used in ImagenFew.
> For instance, we excluded the Weather dataset from evaluation due to the presence of numerous invalid outliers (e.g., -9999), which could negatively impact the fair and accurate evaluation of generation performance.
>
> We use other valid datasets, and strictly follow the official code and training procedures provided by ImagenFew. All experiments are conducted on a local machine with Nvidia V100 GPU and 32 GB memory. The metrics on specific dataset detailed in the appendix align closely with the results reported in ImagenFew. However, due to the exclusion of certain problematic datasets, the average metric results reported in Tables 1 and 2 differ from those in ImagenFew.

---

> > ### Author Rebuttal · Reviewer_b8gK · 2026-04-04
> >
> > Thank you for the clear and constructive rebuttal. The additional clarification on handling variable numbers of variates (W1) is helpful and resolves my concern: using masking and padding is a reasonable and practical solution that aligns with common practice. The analysis of the prototype number K (W2) is also valuable, and the observed non-monotonic behavior provides useful insight into the capacity–generalization tradeoff, strengthening the empirical grounding of the method.
> >
> > The explanation regarding the discrepancy with ImagenFew (W3) is convincing, and it is reassuring that the differences stem from dataset filtering rather than inconsistencies in implementation. Overall, the rebuttal addresses my main questions well and improves both the clarity and credibility of the work. While some minor presentation improvements would still be beneficial, I view the concerns as largely resolved.

---

> > > ### Author Response · Authors · 2026-04-07
> > >
> > > We sincerely appreciate your time and constructive feedback throughout the review process. We are particularly grateful for the encouraging comments and support for our work.
> > >
> > > We are glad that the rebuttal has addressed your concerns and improved the clarity and credibility of the paper. Following your suggestions, we will incorporate additional details on data processing and dataset selection into the revised manuscript to provide a clearer and more comprehensive understanding of TimeMoDE. We will also further refine the writing to improve overall presentation and readability.
> > >
> > > If you have any further questions or concerns, we would like to address them to help resolve any remaining confusion.
> > >
> > > If the revisions meet your expectations, we would greatly appreciate your consideration in supporting our paper and potentially raising the score.

---

### Official Review · Reviewer_i2U6 · 2026-03-09

**Soundness:** 2
**Presentation:** 3
**Significance:** 2
**Originality:** 2
**Overall Recommendation:** 3
**Confidence:** 4

**Summary:**

This paper proposes TimeMoDE, a framework for time series generation in data-scarce settings. It combines Diffusion Transformers with a Mixture-of-Experts architecture to improve domain adaptability and diffusion-stage awareness. The model is pre-trained on large-scale multi-domain datasets to learn both domain-agnostic temporal representations and domain-specific patterns. Domain prompts guide expert assignment for noised tokens, while diffusion timestep signals enable experts to adapt to different denoising stages. Experiments show that TimeMoDE outperforms existing methods across various low-data scenarios, demonstrating strong performance for few-shot time series generation.

**Compliance With Llm Reviewing Policy:**

Affirmed.

**Final Justification:**

Due to the remaining concerns about the novelty and baseline comparison, I decide to maintain my score.

**Key Questions For Authors:**

1. Could the authors report the number of model parameters? It would help clarify whether the performance gains come from the proposed architecture rather than increased parameter size or training data scale. In addition, please provide inference-efficiency comparisons with the baselines. Also, for generation-based metrics, are the reported results computed over multiple samples (e.g., mean or median), and if so, which aggregation strategy is used?
2. How sensitive is the performance to the design of the Domain Prompt? Do different prompt formulations lead to noticeable performance variations? It would also strengthen the evaluation to include statistical significance checks (e.g., reporting mean and standard deviation). Some metrics, such as Predictive Score, appear close to saturation, so a more rigorous statistical evaluation would help better support the conclusions.

**Limitations:**

Yes

**Strengths And Weaknesses:**

### Strengths

1. The paper presents a clear motivation and a strong problem statement.
2. The paper is well presented, with clear explanations and illustrations.

### Weaknesses

1. The proposed method builds on several existing components, including DiT as the diffusion backbone, MoE-style modules (MoDE), domain prompts, and prototypes for routing and transfer. While the integration is well motivated, these components are relatively common in the time-series literature. As a result, the main contribution appears to lie in combining these established modules into a system for cross-domain few-shot diffusion generation, rather than introducing a fundamentally new modeling paradigm or core mechanism. This somewhat limits the perceived originality of the work.
2. The discussion of related work does not fully cover the growing line of research on text-conditional time series generation [1,2,3], which has received increasing attention in recent years. Such approaches condition generation on textual descriptions and can potentially provide domain information. Including these works in the discussion and, if possible, in experimental comparisons would help place the proposed method in a more complete research context.
3. Although the appendix states that there is no overlap between the pretraining and evaluation datasets, some datasets appear closely related. For example, ETTh1 and ETTm1 correspond to different sampling rates of the same underlying time series, which may affect the fairness of evaluation. In addition, some baselines in the main table are not designed for data-scarce generation, so comparing under few-shot settings can also be misleading. As a result, it is unclear whether the reported gains come from the proposed method itself or simply from higher training cost.

[1] VerbalTS: Generating Time Series from Texts. ICML 2025.

[2] T2S: High-resolution Time Series Generation with Text-to-Series Diffusion Models. IJCAI 2025.

[3] BRIDGE: Bootstrapping Text to Control Time-Series Generation via Multi-Agent Iterative Optimization and Diffusion Modeling. ICML 2025.

---

> ### Author Rebuttal · Authors · 2026-03-31
>
> We thank Reviewer i2U6 for detailed and thoughtful review. Below, we try to address each point and will incorporate feedback in revision.
>
> **W1** *TimeMoDE builds on existing components... limits perceived originality*
>
> We would like to clarify and emphasize our novelty.
> The core contribution lies in establishing a strong baseline for cross-domain few-shot time series generation, a meaningful scenario yet largely overlooked in community.
> To tackle data scarcity, we introduce TimeMoDE that goes beyond restrictive assumption commonly adopted in prior work.
> Moreover, we identify unique challenges: (1) indistinguishable noise across domains (2) variation of time series degradation in diffusion, which are tackled by proposed DP-based routing and stage-aware expert.
>
> While some individual components (e.g., DiT) aren't new, they serve as means to realize our contribution rather than constituting core contribution themselves.
> Besides, integrating these paradigms into a coherent framework is non-trivial. It requires careful adaptation and effectively coordinating interaction, which represents pioneering empirical exploration absent in prior work.
>
> **W2** *Related work doesn't fully cover research on text-conditional time series generation [1,2,3]...*
>
> We will include discussion noted by reviewer as follows:
>
> Though text description of time series segment can reflect temporal pattern to some extent, they suffer from drawbacks:
> (1) As presented in [1,2,3], the text typically captures coarse pattern (e.g., time series exhibits an upward trend) while ignoring statistical property (e.g., peak magnitude) meaningful and vary significantly across datasets. It ultimately degrades fidelity of generated sample.
> In addition, modality discrepancy inevitably discards local detail. Especially in few-shot, coarse text struggle to distinguish subtle yet critical temporal feature among limited sample, and homogeneous text conditions result in generation lacking diversity.
> (2) Furthermore, the text construction relies on LLM or agent, where complex prompt and procedure raise additional cost.
>
> **W3.1** *ETTh1 and ETTm1... may affect fairness of evaluation*
>
> We appreciate reviewer for pointing out the issue. We admit that ETTh1 may introduce fairness concern on ETTm1.
> However, ETTh1 (hourly) and ETTm1 (minute-level) exhibit substantially different characteristics under fixed length (e.g. periodicity and short-term dynamic). Therefore, the influence is limited, and fine-tuning on ETTm1 remains challenging.
>
> More importantly, beyond this pair of datasets, our evaluation spans numerous fully independent datasets, providing strong and fair assessment.
>
> **W3.2** *Some baselines are not designed for data-scarce generation...*
>
> As discussed in manuscript, most works are developed under assumption of abundant training data. It's their key limitation and leads to the lack of suitable baselines for cross-domain few-shot generation.
> One of our contributions is moving beyond this assumption to address a more practical and challenging task.
>
> We carefully select representative baselines in recent years.
> While ImagenFew shares a similar target with TimeMoDE, it provides persuasive evaluation.
> We also report full-shot results, which fairly demonstrate the superiority of TimeMoDE.
>
> **Q1** *Could authors report the number of model parameters...*
>
> We report results requested by reviewer in Table (https://anonymous.4open.science/r/ICML_i2U6/Q1.png).
> As TimeMoDE serves as a foundation model for cross-domain modeling, it typically involves more parameters than single-domain model. However, the gain cannot be attributed solely to increased parameter as single-domain model under such scaling tends to suffer from mode collapse and degraded quality.
> Our improvement arise from carefully designed novel architecture, which captures domain-agnostic pattern and domain-specific representation from large-scale datasets with scalability and generalization
>
> All results are computed as the mean over multiple samples for reliability.
>
> **Q2** *How sensitive is performance to the design of Domain Prompt...*
>
> Domain Prompt (DP) aims to effectively capture temporal dependencies and multivariate correlations. We adopt lightweight linear projection on temporal dimension and dynamic convolution on channel dimension for the expressive DP.
> To evaluate sensitivity of the design, we construct variants by removing them respectively.
> The performance degradation indicates the necessity of both information.
> Exploring more diverse DP formulations is an interesting question. We thank reviewer for raising it and highlight it as potential direction.
>
> We report mean and variation range of results over 10 runs in appendix. While Pred. appears close to saturation in a few cases, c-FID and Disc. directly assessing distribution similarity show pronounced performance gap by TimeMoDE. Together with rigorous statistical validation suggested by reviewer, they provide strong support for conclusion.

---

> > ### Author Rebuttal · Reviewer_i2U6 · 2026-04-03
> >
> > Thanks for the authors' response. I still have concerns regarding the novelty of the proposed method and the comprehensiveness of the baseline comparison.
> >
> > Existing work [1] (not discussed or included in the comparison) has already used DiT-based architectures to build time series foundation models that support tasks such as forecasting, imputation, and generation. In addition, MoE-based designs [2,3] are already widely adopted in the time series domain. Therefore, the main contribution of this paper appears to be primarily limited to adapting existing components to the generation setting, particularly through the proposed MoDE Router.
> >
> > At the same time, I do not fully agree with the authors' discussion of text-based generation:
> > - Statistical characteristics can be naturally described through text. In addition, text can also incorporate domain information, morphological patterns, and other contextual signals. Such information is often available at little or no additional cost, for example through statistical summaries, metadata, or lightweight automatic tools.
> > - Text can help constrain the distribution of generated samples, playing a role similar to expert selection. Without direct comparison, the authors' claim that text-based generation reduces fidelity and diversity does not seem sufficiently supported by experimental evidence and is not entirely intuitive.
> >
> > Overall, due to the above concerns, I have decided to maintain my current score.
> >
> > [1] TimeDiT: General-purpose Diffusion Transformers for Time Series Foundation Model.
> >
> > [2] Time-MoE: Billion-Scale Time Series Foundation Models with Mixture of Experts.
> >
> > [3] Moirai-MoE: Empowering Time Series Foundation Models with Sparse Mixture of Experts.

---

> > > ### Author Response · Authors · 2026-04-04
> > >
> > > We sincerely thank the reviewer for the time taken to review our submission and the feedback. We are pleased to further elaborate on the raised concerns.
> > >
> > > **C1** *The novelty of the proposed method and the comprehensiveness of the baseline comparison.*
> > >
> > > We respectfully disagree with the characterization of our work as primarily a combination of existing components. **Our contribution lies not in just an architectural refinement, but in addressing a critical yet largely overlooked challenge**: the severe performance degradation of time series generation in data-scarce regimes.
> > > While DiT backbones and MoE-based designs have been explored in prior work, our method is not a straightforward combination of these techniques. **We leverage them as effective blocks and introduce carefully designed mechanisms tailored to the problem, instead of treating their combination as the source of novelty.**
> > >
> > > To the best of our knowledge, our work is the first to:
> > >
> > > - Identify the unique challenges in cross-domain generative modeling for time series, namely noise indistinguishability and temporal degradation variability, which provides meaningful insights for relevant research.
> > > - Propose a strong and versatile pre-training framework, incorporating MoDE and stage-aware experts, which substantially reduces the performance gap with existing state-of-the-art methods and enables robust transferability.
> > > - Provide an in-depth analysis of the internal mechanisms of TimeMoDE, which offers practical guidance for future architectural design in community.
> > >
> > > We believe these aspects constitute meaningful contributions to the field, as novelty should not be solely defined by introducing entirely new modules, but also by advancing understanding and well-motivated improvements over prior work that enable new capabilities.
> > >
> > > We have incorporated [2,3] into the Related Work section. While the manuscript have included works similar to [1], we thank the reviewer for pointing this out and have further revised the corresponding section to provide a more comprehensive discussion.
> > >
> > > We do not include [1] in the experimental comparison due to the lack of publicly available code, which makes faithful reproduction and fair comparison difficult. If there are additional baselines that are available and particularly critical, we would greatly appreciate the reviewer's suggestions.
> > >
> > > **C2** *Discussion of text-based generation.*
> > >
> > > We thank the reviewer for the insightful and detailed discussion on text-based generation. We appreciate these perspectives and provide the following clarifications:
> > >
> > > - While we agree that statistical characteristics can be described in text, existing text-based time series generation methods mentioned by reviewer [4,5] do not incorporate such detailed statistical information in practice. Instead, they primarily rely on coarse descriptions of temporal patterns as noted in our previous response. More importantly, the main cost does not lie in obtaining numerical statistics, but in constructing high-quality textual descriptions via LLMs or agent-based pipelines. The reviewer’s suggestion of enriching text with statistical summaries is interesting and potentially promising, and we view it as a valuable direction for advancing text-based generation research.
> > >
> > > - Current approaches [4,5] typically condition on text descriptions of individual time series instances, rather than modeling properties at the distribution level. While [6] provides a potential solution through multi-agent collaborative workflows, it incurs substantial computational overhead and is difficult to incorporate within the rebuttal. Also, it falls outside the main scope of our current study. Prior work [7] has discussed inherent limitations of text-based conditioning and reached conclusions consistent with our observations. We agree with the reviewer that additional empirical comparisons would further strengthen this discussion, and we regard it as a valuable evaluation for future investigation.
> > >
> > > Overall, we appreciate the reviewer’s thoughtful discussion, which help us improve the completeness of our Related Work section.
> > >
> > > We sincerely hope that our explanation helps alleviate the concerns and provides a clearer understanding of our work. We would greatly appreciate it if the reviewer could reconsider the evaluation and support our work.
> > >
> > > **References:**
> > >
> > > [1] TimeDiT: General-purpose Diffusion Transformers for Time Series Foundation Model.
> > >
> > > [2] Time-MoE: Billion-Scale Time Series Foundation Models with Mixture of Experts.
> > >
> > > [3] Moirai-MoE: Empowering Time Series Foundation Models with Sparse Mixture of Experts.
> > >
> > > [4] VerbalTS: Generating Time Series from Texts.
> > >
> > > [5] T2S: High-resolution Time Series Generation with Text-to-Series Diffusion Models.
> > >
> > > [6] BRIDGE: Bootstrapping Text to Control Time-Series Generation via Multi-Agent Iterative Optimization and Diffusion Modeling.
> > >
> > > [7] TimeDP: Learning to Generate Multi-Domain Time Series with Domain Prompts.

---

### Official Review · Reviewer_SjuC · 2026-03-12

**Soundness:** 3
**Presentation:** 3
**Significance:** 3
**Originality:** 2
**Overall Recommendation:** 5
**Confidence:** 4

**Summary:**

TimeMoDE is a framework that brings the MoE mechanism into the DiT architecture. It swaps out the standard MLP layers for a Domain Expert Blending module, which uses domain prompts and time-series prototypes to pick the best experts for specific noisy samples. It also includes a stage to sharpen temporal signals using the diffusion process. Because the model is pre-trained on a wide range of datasets, it picks up both general patterns and details specific to certain domains. This allows it to work well with very few samples during fine-tuning, making it a practical option for generating time series data in scenarios where data is hard to come by.

**Compliance With Llm Reviewing Policy:**

Affirmed.

**Final Justification:**

Thank the authors for their detailed response to my concerns. They have not only clearly explained the design details of the classifier guidance module, allowing me to fully understand the core logic of their method, but also proposed an insightful diffusion-based prediction approach for downstream forecasting tasks, supported by convincing and high-quality experimental results. After comprehensive consideration, I have decided to increase my score for this paper.

**Key Questions For Authors:**

1. The model uses both class labels and domain prompts to guide the generation process, but it is not clear how these two types of information interact. Since class labels and domain prompts both carry domain-related data, you should verify if they are redundant or if they actually complement each other. It would be helpful to see if fusing the class labels directly into the routing mechanism could improve how experts are assigned. Most importantly, the necessity of the class labels themselves remains unproven. You should provide results for a version of the model that does not use class labels at all to show if the domain prompts can handle domain differentiation on their own across few-shot and full-shot settings.
2. The paper does not explicitly describe how the model handles a completely new domain that was not part of the pre-training set. If a target domain has no class labels for reference, it is unclear how the domain prompt and prototype matching mechanisms achieve adaptation. Please clarify if you used a training style similar to classifier-free guidance where labels are randomly removed to improve generalization. If you did, please show the results for both conditional and unconditional generation. If you did not use that approach, please explain the specific technical path for handling unknown domains and provide experimental evidence that the model can still generate high-quality data in those scenarios.
3. The current ablation experiments only remove one module at a time, which does not show how the different parts of the architecture work together. Since the synergy between these modules is a major part of your design, testing them in isolation is not enough to prove the framework is solid. You should include combined ablation experiments where multiple modules are removed at the same time. This would help identify the core dependencies between the different parts and show which combinations are most critical for the model to function properly.
4. You have verified that the model can generate time-series data, but the practical value of a generative framework often depends on how well it supports downstream tasks like forecasting, anomaly detection, or imputation. Given that your framework is designed for general domain adaptability and uses a diverse pre-training set, it would be a strong addition to see it applied to these practical scenarios. Extending your experiments to include these tasks would better demonstrate the real-world potential of the model and show that the generated data is useful for more than just looking similar to the training set.
5. The current comparison is limited to four baseline models, which may not be enough to represent the current state of time-series generation. Including more mainstream or recent methods such as TimeGAN, TimeVAE, or DiffTime would make the performance results much more persuasive. When you add these comparisons, please make sure to keep the evaluation metrics and experimental settings consistent. Using statistical analysis to confirm that the performance gaps are significant would also help make your conclusions more credible to the reader.
6. You chose to build your prototypes using noisy samples, which seems counterintuitive since noise can mask the stable features that prototypes are supposed to capture. Original samples without noise would likely reflect the essential characteristics of a domain much more accurately and lead to better expert assignment. Please explain the reasoning behind using noisy samples for this process and provide a comparison to show whether using clean samples would improve or hinder the accuracy of domain recognition.

**Limitations:**

yes

**Strengths And Weaknesses:**

Soundness
The technical logic is solid, using a Diffusion Transformer paired with a MoE to handle cross-domain generation when data is scarce. The way the framework uses domain prompts and prototypes to guide the model at different diffusion steps is a smart approach to making the system aware of the data it's handling. However, there are some gaps in the testing: you didn't show what happens if category labels aren't used, so it’s hard to tell if the domain prompts are doing the heavy lifting or if the labels are carrying the model. There is also no evidence for how the model performs in a completely unknown domain, and the ablation study only looks at one module at a time rather than how they work together. Additionally, building prototypes from noisy samples is an unusual choice that wasn't compared against using clean samples to justify why it works better.
Presentation
The paper follows a standard academic structure and is generally easy to follow. It includes clear figures and helpful hyperparameter tables that are great for reproducibility. That said, some of the specific workflows are a bit blurry, especially the process for generating data in unknown domains and the training mode for scenarios where labels are not available. You could also spend more time explaining the DiT backbone itself and the specific methods used for conditional embedding, as these parts currently feel like a black box. Making these technical details more transparent would make the framework much easier for an expert reader to fully understand and implement.
Significance
This work is very practical because it tackles the common real-world problem of having very little time-series data in sensitive fields like finance or healthcare. Being the first to combine Mixture of Experts with Diffusion for this specific task is a notable milestone that could guide future generative models in this area. However, the actual impact is a bit limited because the paper only evaluates the generation task itself. It does not show if this synthetic data actually helps with practical downstream tasks like forecasting or anomaly detection. Also, since the model performance relies heavily on a massive cross-domain pre-training phase, it might be difficult for users without those huge datasets to get the same level of results.
Originality
TimeMoDE is a clever bit of engineering that moves away from the usual MLP layers in a Diffusion Transformer and uses a Mixture of Experts module instead. The way it uses domain prompts and prototype routing to handle noisy samples is a fresh and effective way to deal with different types of data at once. While the combination of these existing tools is well-designed for few-shot scenarios, the innovation is mostly about architecture and implementation. It does not really offer new theoretical breakthroughs in how Diffusion or Mixture of Experts routing works at a fundamental level, so it is more of a high-end design optimization than a brand-new theory.

---

> ### Author Rebuttal · Authors · 2026-03-30
>
> We thank Reviewer SjuC for the thoughtful review and constructive suggestion. Below, we try to address each point in detail and will revise paper to incorporate feedback. Due to character limitation, we move Tables in the response to anonymous link (https://anonymous.4open.science/r/ICML-Rebuttal-SjuC/README.md), with clear captions to ensure proper correspondence.
>
> **W1+Q1** *You should provide results that does not use class labels at all to show if domain prompts can handle domain differentiation on their own.*
>
> Given the insightful suggestion, we add an experiment by completely removing class labels from TimeMoDE. As shown in Table 1, the variant exhibits a slight performance degradation. However, its performance remains comparable and still surpasses baselines. This indicates that Domain Prompts play a primary role in distinguishing different domains, while class labels provide complementary information that further enhances performance.
>
> **W2+Q2** *There is also no evidence for how model performs in a completely unknown domain. Please explain the specific technical path and provide experimental evidence that the model can still generate high-quality data.*
>
> Our test sets include both domains observed during pre-training and entirely unseen ones. For instance, Mujoco (physics) and AirQuality (nature) used in fine-tuning are completely new domains with no coverage during pre-training.
>
> We do not adopt a classifier-free guidance style when handling unknown domains. Specifically, we extend the one-hot encoding of class labels to accommodate new domains, which helps anchor model to new distribution while maintaining unified input format. The Domain Prompts are then constructed from few samples to assign experts that have handled similar temporal patterns during pre-training. It facilitates model to adapt efficiently to previously unseen domains.
>
> As shown in Table 2 and 3, TimeMoDE consistently generates higher-quality samples, demonstrating its strong generalization capability.
>
> **W3+Q3** *The ablation study only looks at one module at a time rather than how they work together.*
>
> We thank reviewer for the constructive suggestion. While we have conducted extensive single module ablations that demonstrate necessity and effectiveness of each component, we agree that combined ablations could further reveal their interdependencies and strengthen overall justification.
> However, due to time and space constraints, it is challenging to include such additional experiments at this stage. We will incorporate the results in final revision to provide a more comprehensive analysis.
>
> **W4+Q6** *Building prototypes from noisy samples is an unusual choice that wasn't compared against using clean samples.*
>
> We apologize for the ambiguity arising from the possible vague statements. In fact, prototypes in TimeMoDE are learned from clean samples rather than noisy ones. Specifically, prototypes are randomly initialized and then iteratively refined during pre-training by minimizing discrepancy with domain prompts encoded from clean time series, thereby capturing domain-specific characteristics.
>
> The motivation behind our design aligns with reviewer's intuition: we attempt to make prototypes capture essential patterns and stable feature of each domain from original clean data. It facilitates more accurate domain inference and reliable expert assignment.
>
> **W5+Q4** *It does not show if synthetic data actually helps with practical downstream tasks like forecasting or anomaly detection.*
>
> The Predictive Score (Pred.) reported in manuscript reflects the utility of synthetic data for forecasting tasks. Specifically, an LSTM-based model is first trained to predict future time series using sequences of generated past timesteps. The trained model is then evaluated on real data by computing the Mean Absolute Error of forecasting loss. A lower value indicates that the synthetic data is more beneficial for reducing prediction error in downstream tasks.
>
> As shown in Table 4, TimeMoDE achieves lower values compared to baselines in most cases, demonstrating that the synthesized data of TimeMoDE is of higher quality and provides stronger practical utility.
>
> **Q5** *The current comparison is limited ... Including more mainstream methods such as TimeGAN, TimeVAE, or DiffTime would make the performance results much more persuasive.*
>
> We select representative works in recent years as baselines, including ImagenFew (2025), DiffusionTS (2024), ImagenTime (2024), and KoVAE (2024). We do not include TimeGAN (2019) and TimeVAE (2021), as they are relatively outdated and have been verified to be outperformed by our baselines.
>
> Following reviewer's suggestion, we include TimeVAE to make the comparison more comprehensive and persuasive. As shown in Table 5, under consistent settings, TimeMoDE demonstrates a clear performance advantage. We further validate the results using statistical significance tests, which substantiate the superiority of TimeMoDE.

---

> > ### Author Rebuttal · Reviewer_SjuC · 2026-04-03
> >
> > I still have doubts about the generation steps for unknown domains. Although I understand that in the backbone architecture of DiT, the authors' most significant contributions to the generation of new domains are the DP and MoE designs, and the class conditioning of DiT plays an insignificant role. However, for completely unseen samples, what kind of class condition will be input into DiT? Is this condition designed as a one-hot vector? Yet there should be no pre-defined one-hot design for completely unknown samples. Regarding the authors' response that the Pred score can reflect prediction performance, I prefer that the authors evaluate prediction using direct generation results rather than such generation metrics, as this places more emphasis on assessing the quality of generation and the quality of representations.

---

> > > ### Author Response · Authors · 2026-04-07
> > >
> > > We sincerely thank the reviewer for the time and effort devoted to evaluating our work, which helps us further improve the paper. We would like to address the raised concerns to provide a clearer and more comprehensive understanding of our approach.
> > >
> > > **C1** *Doubts about the generation steps for unknown domains.*
> > >
> > > In our design, we adopt the one-hot vector as class condition. Specifically, we initialize the dimensionality of the one-hot vector to be larger than the number of domains observed during pre-training. This allows the condition space to fully cover all known domains while reserving additional positions for potential unseen domains. These reserved class labels are not involved during pre-training and act purely as placeholders to differentiate data sources, without encoding any information from unknown domains.
> > >
> > > During fine-tuning, if the dataset belongs to a known domain, we use the corresponding one-hot vector to guide the model to sample from the learned distribution. In contrast, for completely unseen domains, we activate the reserved positions in the one-hot vector to form a new class label. This indicates that the data originates from a previously unseen distribution while maintaining a consistent input format. Since this label does not carry additional domain-specific information, the model relies on the temporal patterns learned during pre-training to adapt its behavior toward the new target distribution.
> > >
> > > This setting aligns with realistic scenarios where unseen domains lack predefined semantic labels. The model is expected to iteratively refine its parameters associated with the new class label during fine-tuning, thereby forming a useful prior when encountering data from the same domain again.
> > >
> > > We have revised the manuscript to clarify this design in more detail for better understanding. The implementation in submitted code also follows the procedure and provides an intuitive illustration of the mechanism.
> > >
> > > **C2** *Evaluate prediction using direct generation results.*
> > >
> > > We thank the reviewer for this insightful suggestion. While TimeMoDE is primarily designed for generative modeling, we agree that evaluating forecasting performance via direct generation provides a more explicit assessment of both generation quality and the learned representations.
> > >
> > > Following this suggestion, we extend TimeMoDE to a conditional generation setting for forecasting. Specifically, we leverage the pre-trained diffusion model together with classifier guidance to approximately sample from the posterior:
> > > $$
> > > p\left(x_{0: T} \mid y\right)=\prod_{t=1}^{T} p\left(x_{t-1} \mid x_{t}, y\right),
> > > $$
> > > where $x_0$ denotes the observed history (conditioning context), and $y$ represents the prediction values (generation target). The conditional transition can be decomposed as:
> > > $$
> > > p\left(x_{t-1} \mid x_{t}, y\right) \propto p\left(x_{t-1} \mid x_{t}\right) p\left(y \mid x_{t-1}, x_{t}\right).
> > > $$
> > > Baed on Bayes' theorem, we perform gradient update on $x_{t-1}$ using the score function:
> > > $$
> > > \nabla_{x_{t-1}} \log p\left(x_{t-1} \mid x_{t}, y\right)=\nabla_{x_{t-1}} \log p\left(x_{t-1} \mid x_{t}\right)+\nabla_{x_{t-1}} \log p\left(y \mid x_{t-1}\right),
> > > $$
> > > where $\log p(x_{t-1}|x_t)$ is defined by the pre-trained diffusion model. $\log p(y|x_{t-1})$ parametrized by the classifier can be approximated via $\nabla_{x_{t-1}} \log p\left(y \mid x_{0 \mid t-1}\right)$.
> > > Intuitively, this term guides the generated samples toward areas with higher likelihood of classifier, enabling the model to generate future trajectories consistent with the observed history while leveraging learned temporal dependencies.
> > >
> > > As shown in Table 6 and Figure 1 (https://anonymous.4open.science/r/ICML-Rebuttal-SjuC/forecast.pdf), given historical observations as conditions, the samples generated by TimeMoDE closely match the ground-truth future values. This demonstrates that TimeMoDE not only performs well on generative metrics but also generalizes effectively to practical forecasting scenarios, supporting a broader range of downstream tasks.
> > > We appreciate the reviewer’s suggestion, which helps us better highlight the practical value and potential of our framework from an additional perspective.
> > >
> > > We sincerely hope that our detailed explanations have addressed the concerns. If the revisions are satisfactory, we would greatly appreciate the reviewer’s support and consideration in reassessing the score.

---

### Official Review · Reviewer_bftN · 2026-03-16

**Soundness:** 3
**Presentation:** 3
**Significance:** 2
**Originality:** 2
**Overall Recommendation:** 3
**Confidence:** 4

**Summary:**

The authors propose TimeMoDE, a unified framework that combines a Diffusion Transformer(DiT) backbone with a Mixture-of-Domain-Experts (MoDE) architecture. The method is pre-trained on a collection of multi-domain time-series datasets and then fine-tuned on a new low-resource target dataset. Its main design components are domain prompts, which summarize domain-specific information from a small set of available samples, prototype-based routing to assign noisy tokens to relevant experts.

**Compliance With Llm Reviewing Policy:**

Affirmed.

**Key Questions For Authors:**

please refer to above

**Limitations:**

They didn't mention the limitaton of the method.

**Strengths And Weaknesses:**

In general, this paper is well written and easy to follow. This paper address fresh and valid problem, time-series generation under severe data scarcity. Also, the authors evaluate its method on various settings, including few-shot settings using both percentage-based and count-based subsampling, and spans multiple application domains such as forecasting, simulation, classification, and generation. Also, it includes decent amount of ablation study, showing how each component contribute to its strong empirical performance. However, in my viewpoint, the domain prompt is just an summary of a small number of target-domain samples. This is good conditioning techniques but it is much closer to exemplar-based conditioning or learned context pooling than to a strong new mechanism. The paper presents it as a major conceptual contribution, but I think it is not very new. Also, I think this paper adapts existing techniques and engineers well to solve the time-series generation problem under data scarcity. However, while it has its own contribution, the writing seems to overstate the novelty of this method. Also, TimeMoDE is pre-trained on a large multi-domain dataset collection, while baselines are not designed around large-scale cross-domain pretraining. So the comparison can be somewhat bit unfair.

---

> ### Author Rebuttal · Authors · 2026-03-30
>
> We thank Reviewer bftN for the thoughtful review and the recognition. Below, we try to address each point in detail to improve clarity and better support our contributions.
>
> **W1** *The domain prompt is just an summary of a small number of target-domain samples. This is good conditioning techniques but it is much closer to exemplar-based conditioning or learned context pooling than to a strong new mechanism.*
>
> Our main contribution lies in identifying the core challenge that the unified generative models struggle to effectively distinguish noisy signals across datasets, which hinders the accurate inference of target domain. To this end, we introduce the Domain Prompt (DP) mechanism. It is not merely an engineer technique but as a novel approach to address the challenge of domain adaptation. While time series tasks often lack domain-specific prior knowledge, leveraging available target-domain samples as guidance to provide model with domain bias forms a label-free and text-free conditioning mechanism. It differs fundamentally from prior works that rely on predefined one-hot encoding labels which have shown limited effectiveness in many scenarios.
>
> Additionally, we introduce the DP construction protocol and Encoder that adapts to pre-training and fine-tuning stages. While the Encoder may seem less complex, it effectively compresses the temporal dependencies and channel correlations within time series into compact representations, which significantly aids the model in distinguishing domains (Figure 5a) and demonstrates empirical value under few-shot and full-shot settings (Tables 1 and 2).
>
> **W2** *I think this paper adapts existing techniques and engineers well to solve the time-series generation problem under data scarcity. However, while it has its own contribution, the writing seems to overstate the novelty of this method.*
>
> We would like to emphasize the innovative aspects of our work.
> One of the core contributions of our paper is establishing a strong, versatile baseline for time series generation under data scarcity, which is both a realistic and meaningful problem that has been largely overlooked. Existing methods are primarily built on the naive assumption of modeling within a single domain, neglecting the potential of domain-agnostic knowledge. This limitation is particularly evident in scenarios where only limited data is available.
> In this work, we introduce a unified modeling framework to address this issue and identifying critical challenges from two distinct perspectives: (1) the indistinguishable noise between domains, and (2) the variations in time series degradation during the diffusion process.
>
> We acknowledge that some components of our approach, such as diffusion models and Mixture of Experts, are not new. However, integrating these paradigms into a coherent and effective unified framework is non-trivial, which presents unique challenges as outlined above. Our carefully designed DP-based routing mechanism and stage-awared expert networks in TimeMoE represents a pioneering effort that effectively address these challenges.
> Through extensive experimental analysis, our results show the improvements gained from the collaboration among modules, which were not present in prior works. It provides a solid foundation and valuable insights for further related research in the community.
>
> **W3** *TimeMoDE is pre-trained on a large multi-domain dataset collection, while baselines are not designed around large-scale cross-domain pretraining. So the comparison can be somewhat bit unfair.*
>
> As we discuss in manuscript, most existing methods are developed under the assumption of having access to large amounts of data within a single domain and ignore the potential value of leveraging information across multiple domains. It is their key limitation which leads to the lack of suitable baseline and highlights the gap that we aim to address. We have carefully selected related and representative works in recent years as baselines.
>
> While ImagenFew has similar target and adopts the pretraining-finetuning protocol on the same dataset as TimeMoDE, we believe that the experimental results under such setting provide a fair evaluation that demonstrates the superior performance of TimeMoDE and the effectiveness of novel architectural design.

---

> > ### Author Rebuttal · Reviewer_bftN · 2026-04-04
> >
> > I appreciate authors' explanation. Thanks for clarification. But I don't think still this method is novel enough and there can be better approach for ensuring fair comparison

---

> > > ### Author Response · Authors · 2026-04-07
> > >
> > > We sincerely appreciate the reviewer’s time and feedback during the review process. We would like to further clarify and re-emphasize the novelty and contributions of our work, in the hope of providing a clearer understanding of its significance.
> > >
> > > - We focus on a practical yet largely under-explored scenario: how to generate sufficient time series data under scarcity for mitigating performance bias in downstream applications. While previous generative approaches widely rely on the assumption of access to abundant training data within single domain to solve specific tasks, the mismatch between the underlying assumption and real-world constraints highlights a key limitation and gap in existing methods, which our work aims to address.
> > >
> > > - We propose the unified generative framework, TimeMoDE, which explores the potential of foundation models for time series generation. By pre-training on large-scale multi-domain datasets, TimeMoDE extracts both domain-agnostic temporal representations and domain-specific information, thereby improving generalization during fine-tuning. We provide extensive experiment results under few-shot and full-shot settings. The consistent and notable performance gap supported by rigorous statistical validation demonstrates our improvements over existing methods, which suffer from severe performance degradation under low-data regimes. As a result, TimeMoDE establishes a strong baseline and highlights empirical value that inspires future work.
> > >
> > > - We identify two key challenges under the setting, namely noise indistinguishability and degradation variations in time series. Motivated by these findings, we propose: (1) Domain Prompt, which guides the model to infer target domain and route experts for high-fidelity sampling. It differs from prior methods that solely rely on pre-defined class labels, which have been shown to be less effective; and (2) Stage-aware expert, which incorporates lightweight timestep signals into individual expert network to better handle stage-dependent denoising requirements across heterogeneous time series during diffusion process. The novel mechanisms and extensive experimental analysis provide valuable insights for related research.
> > >
> > > We believe that identifying the critical challenges, providing an effective solution, and dedicating considerable effort to investigating the underlying mechanisms together constitute a meaningful and timely contribution to the community.
> > >
> > > If there are specific prior works that the reviewer believes overlap with our contribution, we would be very grateful for further clarification to better position our work. We have carefully selected representative models from recent years with available codes to ensure a fair comparison. If there are specific models or evaluation protocols the reviewer considers important for fair comparison, we would sincerely appreciate the suggestions. If there are any remaining concerns, we are also willing to address them in detail to further improve the clarity of our work.
> > >
> > > We genuinely hope that our clarification helps alleviate the reviewer’s concerns and provides a clearer understanding of our contributions. If all concerns are satisfactorily resolved, we would greatly appreciate the reviewer’s consideration in reassessing the score and supporting our paper.

---

### Decision · Program_Chairs · 2026-04-30

**Decision:**

Accept (regular)

**Comment:**

This submission presents a split that reflects a genuine borderline case. The two negative reviewers (bftN, i2U6; both confidence 4) raise legitimate novelty concerns — the individual components are established, and recent works (TimeDiT, Time-MoE, Moirai-MoE) already explore DiT and MoE in the time series domain. The two positive reviewers (SjuC at 5, b8gK at 4; confidences 4 and 5 respectively) view the specific integration for few-shot generation as a meaningful contribution with strong empirical backing. The rebuttal was substantive and reviewer SjuC raised their score after new forecasting experiments, and b8gK considers concerns "largely resolved." Therefore I am leaning towards a borderline accept.